# An anti-CRISPR that represses its own transcription while blocking Cas9-target DNA binding

Xieshuting Deng[1,2], Wei Sun [1], Xueyan Li [1,2], Jiuyu Wang [1], Zhi Cheng [1,2], Gang Sheng[1,2] & Yanli Wang [1,2] ✉

AcrIIA15 is an anti-CRISPR (Acr) protein that inhibits *Staphylococcus aureus* Cas9 (SaCas9). Although previous studies suggested it has dual functions, the structural and biochemical basis for its two activities remains unclear. Here, we determined the cryo-EM structure of AcrIIA15 in complex with SaCas9-sgRNA to reveal the inhibitory mechanism of the Acr's C-terminal domain (CTD) in mimicking dsDNA to block protospacer adjacent motif (PAM) recognition. For the N-terminal domain (NTD), our crystal structures of the AcrIIA15-promoter DNA show that AcrIIA15 dimerizes through its NTD to recognize double-stranded (ds) DNA. Further, AcrIIA15 can simultaneously bind to both SaCas9-sgRNA and promoter DNA, creating a supercomplex of two Cas9s bound to two CTDs converging on a dimer of the NTD bound to a dsDNA. These findings shed light on AcrIIA15's inhibitory mechanisms and its autoregulation of transcription, enhancing our understanding of phage-host interactions and CRISPR defense.

CRISPR-Cas systems are adaptive immune systems found in bacteria or archaea that protect against the re-invasion of mobile genetic elements (MGEs), such as phages and plasmids[1–4]. In response, MGEs have evolved anti-CRISPR (Acr) proteins to evade the host's immune defenses[5,6]. Acrs are versatile inhibitors that are typically small and exhibit low homology to other known protein families in both their protein sequences and three-dimensional structures[7,8]. Acrs subvert CRISPR-systems through various strategies, including promoting Cas protein degradation[9], blocking crRNA loading[10,11], preventing DNA recognition[12–15], and obstructing the activation of the Cas protein by preventing the conformational changes of the surveillance complex[16–19].

Acrs are frequently encoded upstream of conserved anti-CRISPR-associated (*aca*) genes within the same operon, allowing *aca* genes to serve as valuable markers for identifying *acr* genes[20–22]. Notably, all known Aca proteins share a common helix-turn-helix (HTH) DNA-binding motif. By inserting into the major groove of the *acr-aca* operon promoter sequence, the HTH motif of Acas act as transcriptional repressors to inhibit *acr* gene transcription. This repression is crucial

to prevent excessive Acr expression, which can be detrimental to the phage life cycle[23]. This negative-feedback regulation effectively safeguards phage genomes against CRISPR-Cas systems during early infection stages, while avoiding premature bacterial lysis caused by excessive Acr expression in later stages.

In contrast to the typical arrangement where *acr* and *aca* genes are separate within the same operon, certain Acrs are fusion proteins that combine both canonical Acr and Aca components: one domain responsible for inhibiting the CRISPR-Cas system, and the other domain containing an HTH motif. Several such bi-functional Acrs have been identified, including AcrIIA1[9,24,25], AcrIIA6[16,26], AcrIIA13-15[27], and AcrIF24[28–30].

AcrIIA13, AcrIIA14, and AcrIIA15 were identified from *Staphylococcus* genomes and able to repress the DNA cleavage activities of SaCas9. They all have an HTH motif at their N-terminal end[27]. AcrIIA13 has been shown to inhibit SaCas9 by preventing target DNA loading[27]. AcrIIA14, on the other hand, can bind to the HNH domain of Cas9, blocking its allosteric activation[31]. Previous studies suggest that AcrIIA15 may prevent sgRNA loading and therefore formation of

[1]Key Laboratory of RNA Science and Engineering, Institute of Biophysics, Chinese Academy of Sciences, Beijing 100101, China. [2]College of Life Sciences, University of Chinese Academy of Sciences, Beijing 100049, China. ✉e-mail: ylwang@ibp.ac.cn

the Cas9 surveillance complex[27]; however, the precise molecular mechanism of AcrIIA15 remains elusive. It also remains unclear how the Acr and Aca domains work together in fusion anti-CRISPRs.

In this work, we determine the structures of AcrIIA15 in complex with SaCas9-sgRNA and/or in complex with *acr* operon promoter DNA. Our structural and biochemical analyses reveal that, contrary to

previous findings, AcrIIA15[CTD] employs a PAM-mimic strategy to block Cas9 from binding target DNA, similar to several other characterized Acrs. We find that AcrIIA15 forms a homodimer through its N-terminal HTH-containing domain and specifically binds to inverted repeats in the DNA upstream of the *acr* gene. Furthermore, our cryo-EM structure of the Cas9-sgRNA-AcrIIA15-DNA quaternary complex confirms that AcrIIA15 dimer interacts with DNA through its N-terminal domain, and with two SaCas9-sgRNA complexes via its C-terminal domain. Our studies provide significant insights into the molecular mechanisms of fusion Acrs that are able to inhibit Cas9 and autoregulate their own transcription.

## Results

### The C-terminal domain of AcrIIA15 inhibits the activity of SaCas9

The AcrIIA15 protein, consisting of 170 amino acids, is comprised of two domains: an N-terminal domain (NTD) from residues 1 to 56, which contains a HTH motif resembling a transcriptional repressor, and a C-terminal domain (CTD) from residues 57 to 170. AcrIIA15 has been shown to inhibit the cleavage activity of SaCas9[27]. In order to isolate the Cas9 inhibitory activity to a specific domain, we first performed in vitro cleavage assays for SaCas9, *Streptococcus pyogenes* Cas9 (SpCas9), and *Neisseria meningitidis* Cas9 ortholog 1 (Nme1Cas9) in the presence of full-length AcrIIA15 or AcrIIA15[CTD] (Fig. 1a and Supplementary Fig. 1). The results demonstrated that both full-length AcrIIA15 and AcrIIA15[CTD] strongly suppress the DNA cleavage activity of SaCas9, while not affecting the activities of SpCas9 and Nme1Cas9. This indicates that AcrIIA15 specifically inhibits SaCas9 activity, and this inhibitory function resides solely in the CTD of AcrIIA15. Importantly, the order of incubation of full-length AcrIIA15 or AcrIIA15[CTD] with SaCas9 and the single-guide RNA (sgRNA) did not affect its inhibitory function, suggesting that AcrIIA15 does not impact the formation of the SaCas9-sgRNA complex (Supplementary Fig. 2). Our size-exclusion chromatography (SEC) tests further revealed that AcrIIA15[CTD] can bind to SaCas9 in both the apo form and sgRNA-bound state, but not the DNA-bound state (Fig. 1b). Therefore, AcrIIA15[CTD] will be used to investigate the Cas9-inhibitory mechanism of AcrIIA15 hereafter.

### Overall structure of SaCas9-sgRNA-AcrIIA15[CTD] ternary complex

To elucidate the molecular mechanisms underlying AcrIIA15 inhibition, we solved the cryo-EM structure of SaCas9-sgRNA-AcrIIA15[CTD] ternary complex at a resolution of 3.31 Å (Fig. 2a–c). In the ternary complex, one AcrIIA15[CTD] molecule binds to one SaCas9-sgRNA complex. AcrIIA15[CTD] is situated in the cleft between the PAM interacting (PI) and wedge (WED) domains of SaCas9. Notably, the PAM region of the target DNA would also be located within this same cleft between the PI and WED domains of SaCas9 (Fig. 2d). This observation suggests that AcrIIA15[CTD] competes with the PAM for the binding site on SaCas9.

AcrIIA15[CTD] is composed of five α-helices and a β-sheet consisting of three β-strands (Fig. 2e). DALI server search shows no known structure of similar folds to AcrIIA15[CTD]. The surface of AcrIIA15[CTD] carries a significant negative charge and resembles the size and shape of conventional B-form dsDNA (Fig. 2e). This structural resemblance is reminiscent of other Acr proteins, such as AcrIIA2, AcrIIA4 and AcrIIC5, which are structural mimics of dsDNA that specifically inhibit type II CRISPR-Cas9 systems by competing for the target DNA binding site[12–15,32,33]. To directly test whether AcrIIA15[CTD] prevents the SaCas9-sgRNA effector complex from binding to target DNA, we performed electrophoretic mobility shift assays (EMSAs) in the presence of AcrIIA15[CTD], finding that pre-incubation of AcrIIA15[CTD] with the SaCas9-sgRNA complex hindered the binding of target dsDNA with a canonical 5′-N₂GAAT-3′ PAM sequence (Supplementary Fig. 3). Taken together, our findings show that AcrIIA15[CTD] inhibits the DNA cleavage activity of SaCas9 by preventing it from binding to target DNA.

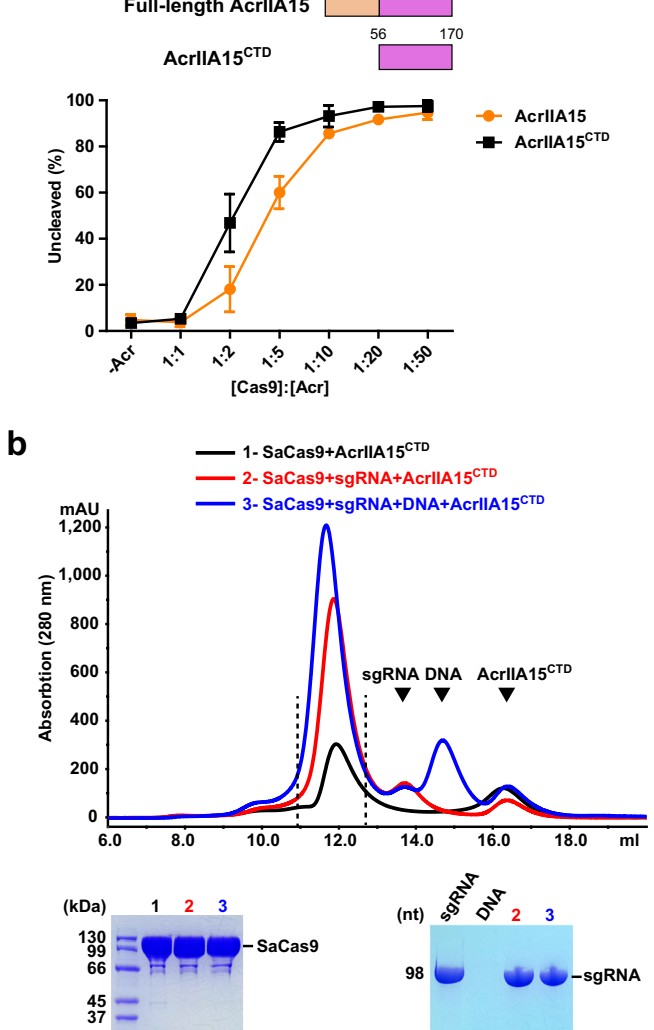

**Fig. 1 | The C-terminal domain of AcrIIA15 binds to SaCas9 to inhibit its cleavage activity. a** In vitro cleavage assay of a target linear plasmid substrate by SaCas9-sgRNA with or without full-length AcrIIA15 or the C-terminal domain of AcrIIA15 (AcrIIA15[CTD]). Sample separation was achieved using agarose gels stained with ethidium bromide. Finally, gels were subjected to statistical analyses (mean ± SD, *n* = 3 independent experiments). **b** Detection of interactions between AcrIIA15[CTD] and SaCas9 in apo, sgRNA-bound, and DNA-bound forms by size-exclusion chromatography (SEC). The molar ratio of SaCas9, sgRNA, DNA and AcrIIA15[CTD] is 1:1.1:1.2:2. Samples from the major peaks (highlighted with vertical black dashed lines) were subjected to SDS-PAGE and Urea-PAGE detection (bottom) stained with Coomassie blue and toluidine blue, respectively. Excess sgRNA, DNA and CTD are indicated by black triangles. Source data are provided as a Source Data file.

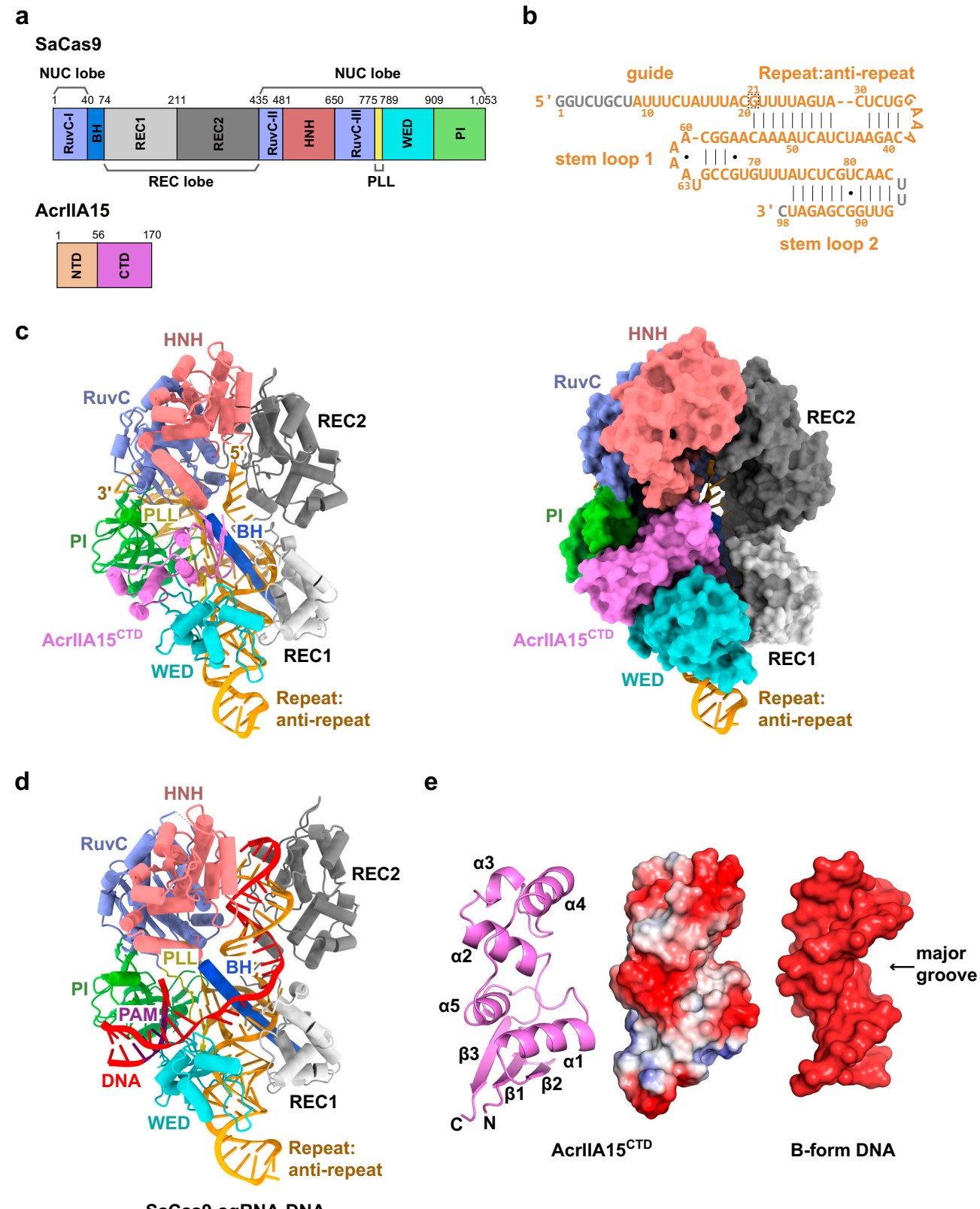

**Fig. 2 | Cryo-EM structure of the SaCas9-sgRNA-AcrIIA15CTD ternary complex.** **a** Domain architecture of SaCas9 and AcrIIA15. **b** Single-guide RNA (sgRNA) of SaCas9. Nucleotides that cannot be traced in our structure are colored in gray. The first nucleotide of the repeat region is highlighted with a dot-line box. **c** Overall structure of the SaCas9-sgRNA-AcrIIA15CTD ternary complex. Domains are colored according to Fig. 2a. Cartoon and surface representations are displayed side by side. **d** Overall structure of the SaCas9-sgRNA-DNA ternary complex (PDB: 5AXW). **e** Cartoon representation of AcrIIA15CTD (left panel) and its electrostatic potential surface representation (middle panel) compared to a surface representation of a typical 20-mer B-form DNA (right panel, PDB: 355D).

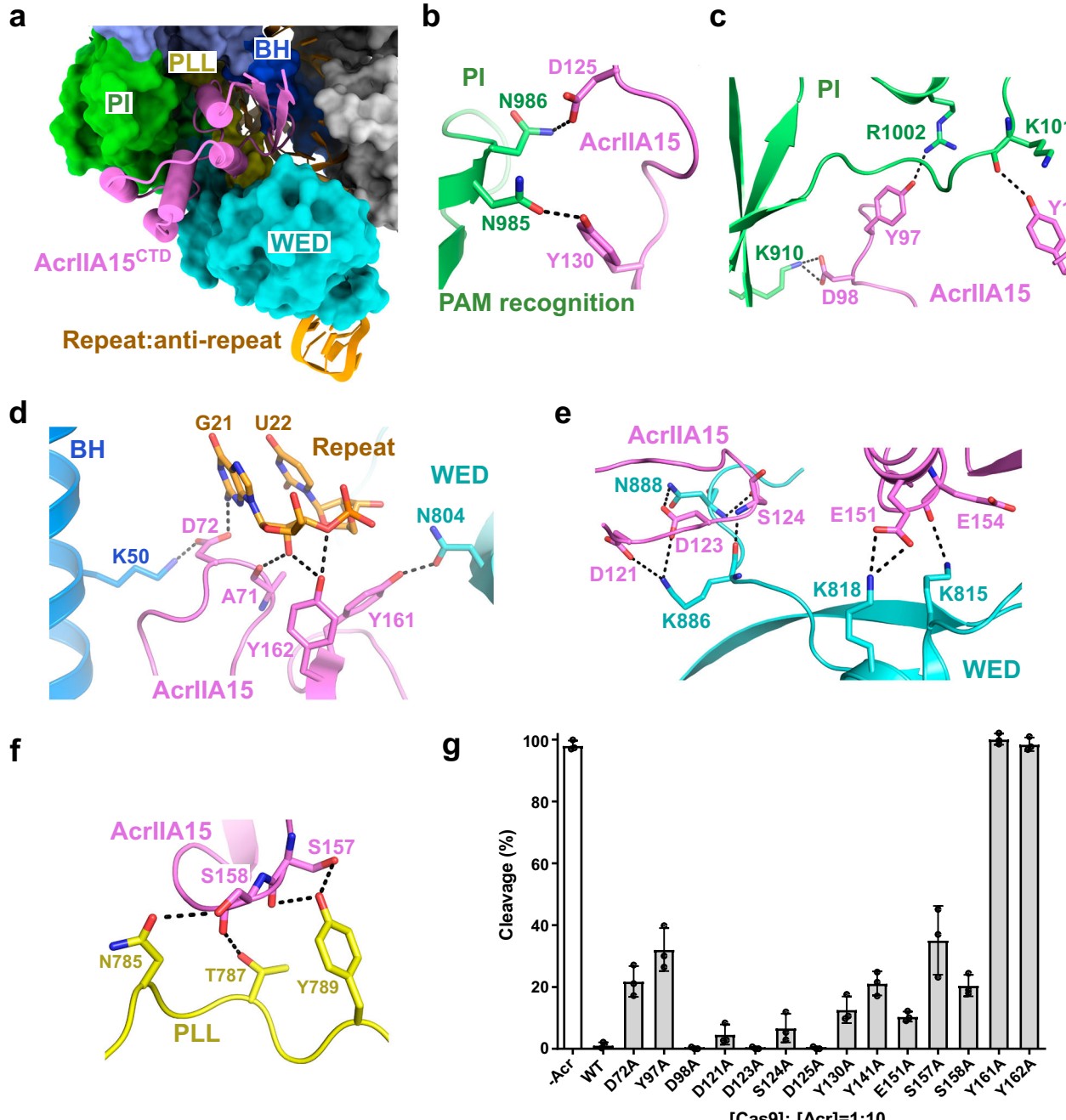

**Fig. 3 | Interactions between AcrIIA15^CTD and the SaCas9-sgRNA binary complex. a** AcrIIA15^CTD located in the crevice between PI and WED domains. **b**, **c** Interactions between AcrIIA15^CTD (magenta) and the PI domain (green). **d**–**f** Interactions between AcrIIA15^CTD (magenta) and sgRNA (orange), BH (blue), PLL (yellow) and WED (cyan) domains. **g** Characterization of mutations in key residues of AcrIIA15^CTD assessed by an in vitro SaCas9-sgRNA cleavage assay using a linear plasmid substrate. The ratio of Cas9 to Acr is 1:10, and the reaction concentration of the enzymatic complex is 100 nM. The ratio and concentration were kept constant for each mutant. Samples were run on 1% agarose gels and stained with ethidium bromide for visualization. Afterwards, the gels were subjected to statistical analyses (mean ± SD, *n* = 3 independent experiments). Source data are provided as a Source Data file.

## AcrIIA15^CTD is positioned in the PAM binding pocket

In our structure, the AcrIIA15^CTD protein is positioned within the crevice and sandwiched by the PI and WED domains of the SaCas9 protein. It interacts with multiple regions of SaCas9, including the PI, WED, bridge helix (BH), and phosphate lock loop (PLL) (Fig. 3a). Specifically, the PAM recognition residues of SaCas9, N985 and N986, form hydrogen bonds with AcrIIA15 residues Y130 and D125, respectively (Fig. 3b), showing that AcrIIA15^CTD binding directly blocks the PAM recognition sites in SaCas9. Furthermore, amino acids D98, Y97, and Y141 of AcrIIA15^CTD interact with amino acids K910, R1002, and K1011 in the PI domain, respectively (Fig. 3c). Additionally, extensive polar interactions between AcrIIA15 and the WED domain are observed (Fig. 3d, e). For the PLL, residues N785, T787 and Y789 are anchored by residues S157 and S158 of AcrIIA15^CTD (Fig. 3f).

The sgRNA also plays a crucial role in stabilizing AcrIIA15^CTD. Specifically, the sidechain of D72 in AcrIIA15 forms hydrogen bonds with residue K50 of SaCas9, as well as with the base of G21 (Fig. 3d), which is the first nucleotide of the repeat region in the sgRNA (Fig. 2b). Furthermore, the phosphate backbone of the repeat region of sgRNA forms additional hydrogen bonds with residues A71 and Y162 of AcrIIA15.

To evaluate the importance of specific residues in AcrIIA15[CTD] for its inhibitory activity, we performed in vitro DNA cleavage assays using SaCas9 and various point mutants of AcrIIA15[CTD] (Fig. 3g). Alanine substitutions for residues Y97 and Y141, which interact with the PI domain of SaCas9, significantly reduced the inhibitory activity of AcrIIA15[CTD]. Mutations of residues D72, S157 and S158, involved in interactions with the BH and PLL of SaCas9 and sgRNA, also markedly decreased the inhibitory activity of AcrIIA15[CTD]. Notably, mutations of AcrIIA15 residues Y161 and Y162, interacting with the sgRNA and the WED domain, completely abolished AcrIIA15[CTD]'s inhibitory activity. In contrast, alanine substitutions of other residues, including D98, D121, D123, S124, D125, Y130, and E151, had no or only a slight impact on inhibition potency. Additionally, circular dichroism (CD) spectroscopy of the seven mutants with reduced inhibition confirmed proper folding of these mutants (Supplementary Fig. 4a, b). These findings highlight the crucial role of these residues in the inhibitory function of AcrIIA15. Moreover, SaCas9 residues interacting with key residues of AcrIIA15 are not conserved in SpCas9 and Nme1Cas9 orthologs, which can explain the inhibition specificity of AcrIIA15 (Supplementary Fig. 5).

Given that both Y161A and Y162A mutants of AcrIIA15 lost their inhibitory activity, we conducted SEC experiments to evaluate whether these mutants retain the ability to bind to the SaCas9-sgRNA effector complex. The Y161A mutant no longer binds to SaCas9-sgRNA (Supplementary Fig. 6a), while the Y162A mutant exhibits weaker binding to SaCas9-sgRNA compared to the wildtype and is easily displaced by the addition of sgRNA-complementary target dsDNA to the mixture (Supplementary Fig. 6b, c). These findings suggest that the interactions between SaCas9-sgRNA effector and Y161 and Y162 are essential for AcrIIA15 inhibitory activity.

## The HTH motif of AcrIIA15 binds to Its promoter region

The NTD of AcrIIA15 protein contains a conserved HTH motif, and previous work has shown that the HTH motif acts as a transcriptional repressor by binding to the promoter region of the *acrIIA15* gene itself, acting similar to Aca proteins in other systems[27]. The promoter region of AcrIIA15 consists of two inverted repeat (IR) regions, namely IR1 and IR2, which are known to be important for Aca1 binding to Acr promoter DNA in *Pseudomonas* phages[23]. IR1 is positioned upstream of the (−35) region, while IR2 is located between the (−35) and (−10) elements. To determine the specific DNA sequence that AcrIIA15 may bind to, we conducted EMSA experiments using full-length AcrIIA15 or the isolated NTD and CTD domains paired with dsDNA containing IR1, IR2, or both IRs (IR1-IR2). We found that the full-length AcrIIA15 exhibits high binding affinity to IR1, IR2 and the entire IR1-IR2, at a nanomolar level (Fig. 4a and Supplementary Fig. 7a). SEC assay showed that AcrIIA15 binds to IR1-IR2 dsDNA sequence at a molar ratio of 4 to 1 (Supplementary Fig. 7b), suggesting that two AcrIIA15 dimers bind to one IR1-IR2 DNA simultaneously. The HTH-containing NTD domain alone also binds to IRs with high affinity. In contrast, the AcrIIA15[CTD] was unable to bind to the IRs, even at high molar ratios (Supplementary Fig. 7c). These results provide further confirmation that the HTH motif in AcrIIA15 is responsible for mediating its interaction with the IRs, while the CTD is dispensable for DNA binding. Considering the sequence similarity and consistent behavior of IR1 and IR2 during the EMSA experiments, we selected IR1 for further investigation, and consider the findings to be likely representative of both IR sequences.

To determine the minimal length of IR1 required for AcrIIA15 binding, we performed a SEC binding assay using two variants of IR1 with different lengths (Supplementary Fig. 8a). We observed that AcrIIA15 was unable to stably bind to DNA truncated by four base pairs from both ends. However, the addition of a single nucleotide to the 5′-ends of both strands efficiently restored the binding of AcrIIA15 (Supplementary Fig. 8b). This observation suggests that the central 20 base pairs within IR1 are critical for AcrIIA15 binding. Based on these findings, we utilized this specific DNA sequence with one base-pair

overhangs to form the AcrIIA15-DNA complex for subsequent crystallization experiments.

## Crystal structure of the AcrIIA15-DNA complex

To investigate how AcrIIA15 recognizes promoter DNA in a sequence-specific manner, we determined the crystal structures of the full-length AcrIIA15 or the NTD domain in complex with IR1 DNA at resolutions of 3.15 Å and 3.10 Å, respectively (Fig. 4b, c). Resembling a canonical HTH-containing DNA-binding protein, our structural analysis revealed that two AcrIIA15 molecules form a homodimer that binds to a single IR1 dsDNA molecule using its NTD. Each HTH motif of an AcrIIA15 dimer binds to one copy of the IR1 inverted-repeat sequence in a similar manner.

The DNA is positioned on the positively-charged surface created by the two HTH domains within the dimer (Fig. 4d). Importantly, the DNA bound to the AcrIIA15 dimer exhibits a bending angle of approximately 30° compared to typical dsDNA conformations (Fig. 4e and Supplementary Fig. 9a). This observation suggests that the binding of AcrIIA15 induces a conformational change in the DNA, highlighting the specific recognition of the promoter DNA sequence by AcrIIA15. Similar DNA bending was also observed when the DNA interacts with HTH from AcrIF24 or Aca1 (Supplementary Fig. 9b), suggesting that the HTH binding affects DNA conformation.

AcrIIA15[NTD] consists of four α-helices, with helices α2 and α3 together forming the HTH motif. In the AcrIIA15-DNA complex, the HTH motif and the subsequent loop establish extensive interactions with the phosphate backbone of the DNA (Fig. 5a). Specifically, helix α3 of the HTH domain inserts into the major groove of the DNA and forms three base contacts with two base-pairs in each IR sequence (Fig. 5b, c). Specifically, amino acids K31 and S25 form hydrogen bonds with the bases of the G6-C6′ pair. In addition, the sidechain of Q26 forms hydrogen bonds with the base of A4 (Fig. 5a, c). The HTH motif in the other monomer of the AcrIIA15 dimer interacts with the second inverted-repeat sequence in the IR1 DNA in a similar manner, and the nucleotides involved in these base contacts are conserved in both IR1 and IR2 (Fig. 5d), providing an explanation for the ability of AcrIIA15 to bind to both inverted repeats with similar affinity.

We conducted EMSA experiments using DNA variants to assess the significance of these base contacts in its interaction with AcrIIA15. Mutations that replaced the A4:T4′ base pair with G:C or the G6:C6′ base pair with T:A in both inverted repeats significantly reduced the binding affinity between AcrIIA15 and DNA. Combining these two mutations in both base pairs completely abolished the interaction (Fig. 5e). Subsequently, we introduced alanine substitutions for base-contact amino acids and carried out EMSAs (Fig. 5f). Alanine substitutions at S25 and Q26 moderately reduced the binding affinity, and the single mutation K31A completely disrupted the interactions between the HTH and DNA. Importantly, these AcrIIA15 mutants fold properly, as confirmed by CD spectroscopy (Supplementary Fig. 4c). These findings show that AcrIIA15[NTD] sequence-specifically recognizes the IR sequences through the establishment of base contacts between helix α3 and DNA.

In addition to the base contacts, the HTH motif of the NTD domain stabilizes the backbone of both DNA strands. The positively-charged surface of the NTD allows for binding to the negatively-charged DNA molecule (Fig. 4d). Specifically, amino acids S14, S15, N16, S30, and N34 establish interactions with the phosphate backbone of one DNA strand adjacent to the 5′-end (Fig. 5a and Supplementary Fig. 9c). Mutating N16 and S30 moderately reduces their DNA binding ability, while S14 and N34 mutations cause a slight decrease (Supplementary Fig. 9d). Additionally, residues K37, T43, and S46 interact with the phosphate groups of nucleotides in the middle spacing regions (Fig. 5a and Supplementary Fig. 9e). Furthermore, K37A fails to bind to DNA, and mutations of T43 and S46 considerably diminish their DNA binding (Supplementary Fig. 9f), indicating the essential nature of these interactions for the NTD domain's binding to DNA.

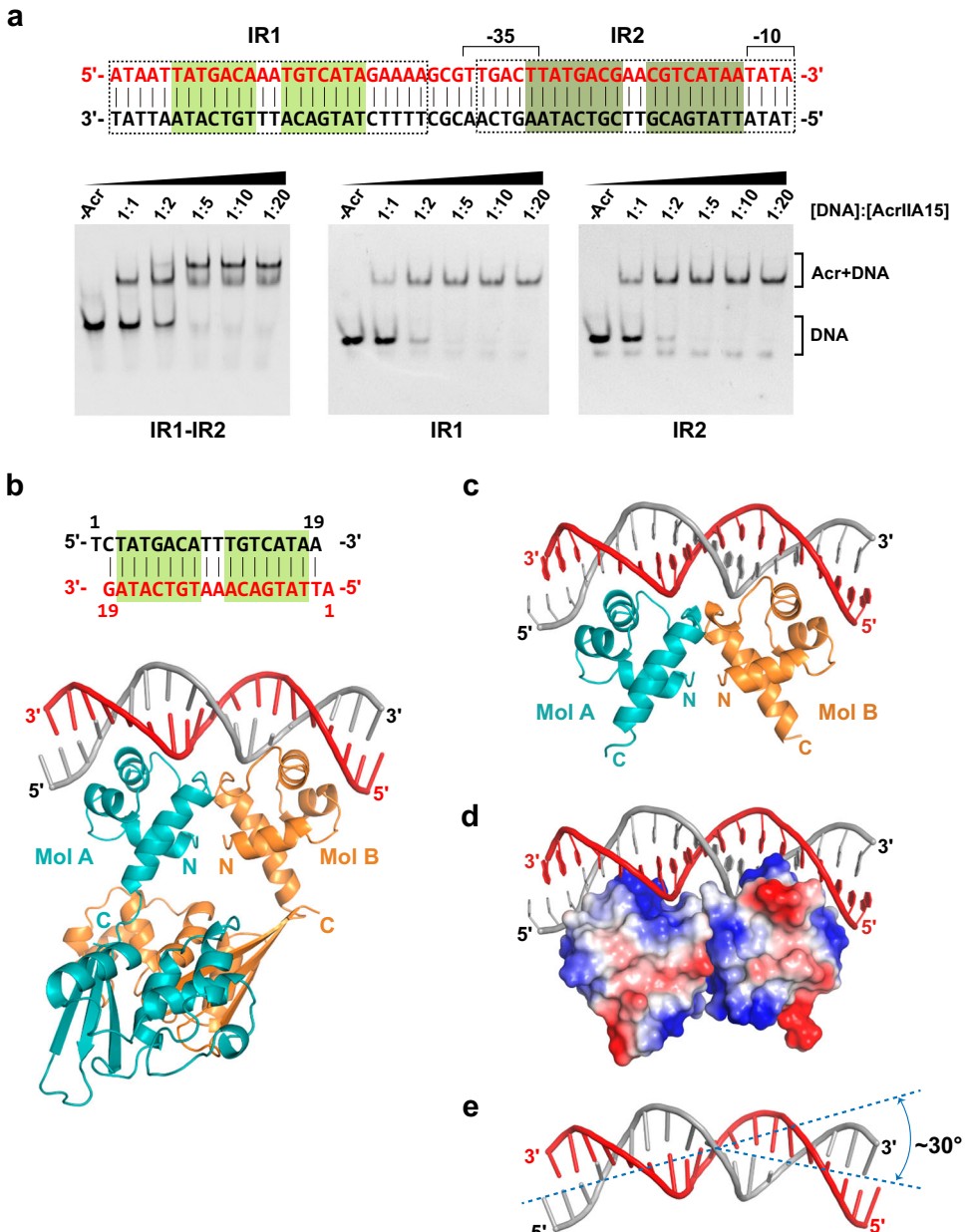

**Fig. 4 | Overall structures of the AcrIIA15-DNA complex. a** EMSA using dsDNA comprised of inverted repeat 1 (IR1), inverted repeat 2 (IR2), or both them (IR1-IR2). Various ratios of DNA to Acr were tested, as indicated across the top of the gels. Samples were loaded onto 5% native gels and stained with ethidium bromide for visualization. **b** Crystal structure of the AcrIIA15-DNA complex. A schematic of the DNA substrate used is depicted at the top. **c** Crystal structure of the AcrIIA15^NTD- DNA complex. **d** Electrostatic surface potential of AcrIIA15^NTD showing a positively-charged surface of AcrIIA15^NTD docked by inverted-repeat DNA. **e** Isolation of the DNA trace from the AcrIIA15-DNA structure. A 30° bending of dsDNA was induced by Acr binding. Source data are provided as a Source Data file.

Next, we conducted a GFP reporter assay to investigate whether AcrIIA15 could repress the transcription of the *sfgfp* gene controlled by the AcrIIA15 promoter in vivo. In the absence of AcrIIA15, sfGFP was expressed normally under the control of the AcrIIA15 promoter. However, in the presence of AcrIIA15, the expression of sfGFP was inhibited (Fig. 5g). Additionally, the K31A mutant, which cannot bind to DNA (Fig. 5f), partially relieved this inhibition (Fig. 5g). These results indicate that AcrIIA15 is able to inhibit transcription under its own promoter.

**Dimerization of AcrIIA15 is essential for binding DNA**
The bending of dsDNA upon binding to AcrIIA15 prompted us to investigate whether AcrIIA15 undergoes conformational changes upon binding promoter DNA. To address this question, we obtained the crystal structure of free AcrIIA15 at a resolution of 2.34 Å (Fig. 6a). We find that full-length AcrIIA15 exists as a dimer, similar to how it binds to DNA. Notably, the NTDs and the CTDs within the dimer share similar structures to what we observed in the DNA-bound and Cas9-bound structures, respectively. However, due to the different relative positions of the CTD and NTD structural domains in the two monomers, the structures of the two monomers in the same dimer are different (Fig. 6b). Structural superposition of the AcrIIA15 structures in the free and DNA-bound states showed that they have similar structures (Supplementary Fig. 10a), suggesting that DNA binding does not induce significant structural rearrangement in AcrIIA15.

The crystal structure and biochemical assays in solution reveal that the full-length AcrIIA15 adopts a dimeric form, while its CTD alone remains as a monomer in solution (Supplementary Fig. 10b).

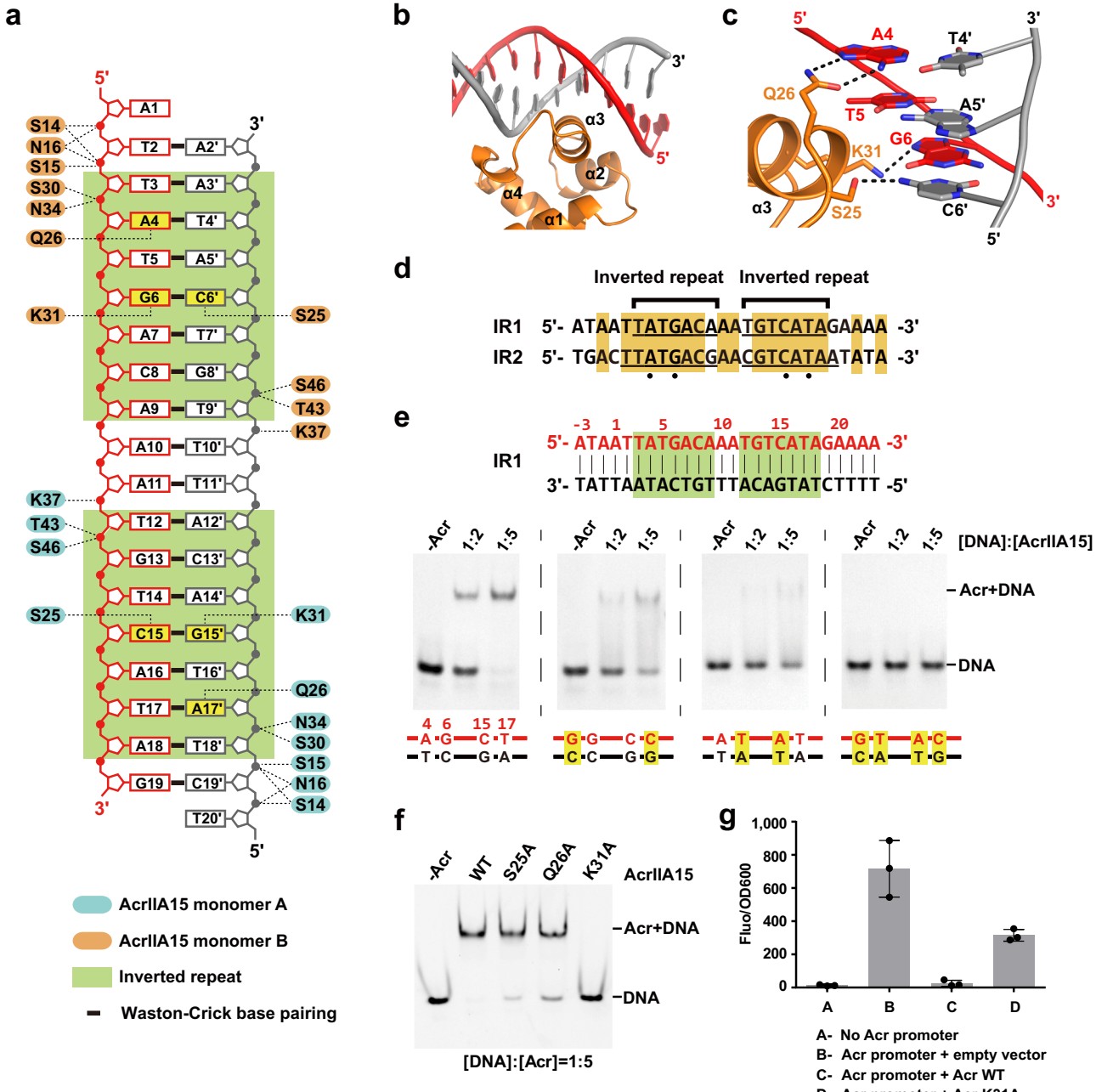

**Fig. 5 | Interactions between AcrIIA15^NTD and dsDNA in the AcrIIA15^NTD-DNA complex. a** Schematic of DNA recognition by AcrIIA15^NTD. Residues are color-coded as in Fig. 4c. Bases of nucleotides involved in base-specific interactions are colored in yellow. **b**, Zoom-in of the HTH domain inserting into the major groove of the DNA. **c**, Illustration of the base-specific interactions outlined in Fig. 5a. **d** Sequence alignment between IR1 and IR2. Orange shading indicates matching nucleotides. The inverted repeats are underlined and the black dots illustrate the nucleotides involved in base-specific interactions. **e** EMSAs evaluating the ability of full-length AcrIIA15 to bind to various indicated mutants of the IR1 DNA sequence. **f**, EMSA characterizing the DNA-binding ability of mutations in key residues of full-length

AcrIIA15 by adding Cy3-labeled IR1 dsDNA and proteins at a molar ratio of 1:5. EMSA samples were separated by 5% native gels and visualized using a FluorChem system. **g**, In vivo transcriptional repression assay by AcrIIA15 using a Superfolder (sf) GFP reporter. The *acrIIA15* promoter region was cloned upstream of a promoterless *sfgfp* expression vector, and a second vector empty or expressing wild-type AcrIIA15 or K31A mutant was co-transformed into *E. coli* BL21 (DE3) strains. Measured fluorescence of sfGFP normalized by bacterial optical density at 600 nm was calculated. Data are mean ± SD (*n* = 3 independent experiments). Source data are provided as a Source Data file.

This finding suggests that the dimerization of AcrIIA15 relies on the HTH motif-containing NTD domain, consistent with our observations from the DNA-bound structure we obtained: the interaction between two AcrIIA15 molecules primarily takes place through their HTH domains (Fig. 6c). The main dimer interface is formed by the loop connecting helices 3 and 4 within the HTH domain, which involves multiple hydrogen bonds and hydrophobic interactions. Residues

G40 and L42 of one NTD form mainchain interactions with residues L44 and N45 of the other NTD. A hydrophobic core is composed of specific amino acids, namely L44, I5, L8, L9, L32, L39, L42, and L50, from both monomers (Supplementary Fig. 10c). Additionally, helix α1 contributes to dimer formation, with residue R2 from one monomer establishing hydrogen bonds with the sidechains of N45 and E48 from the opposite monomer. We also see that interactions between

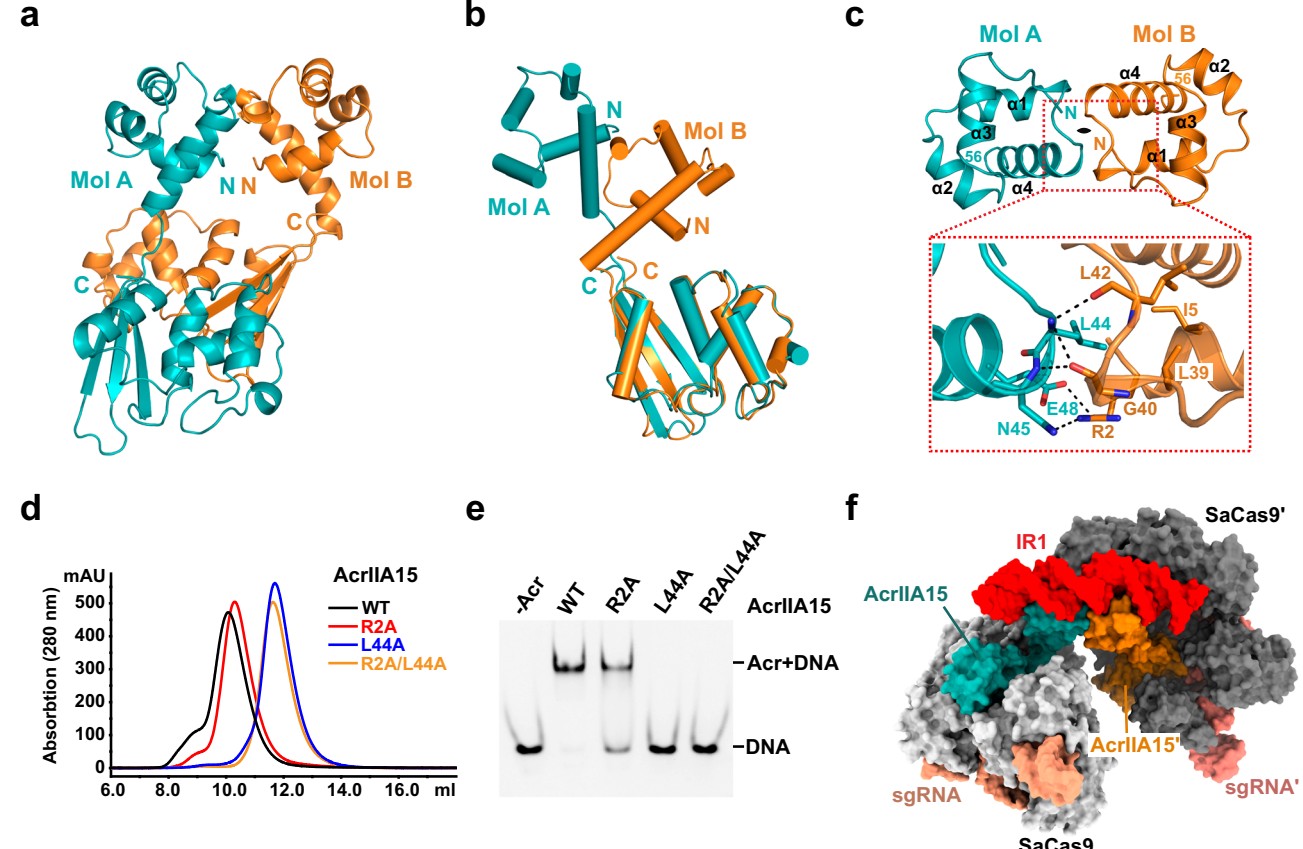

**Fig. 6 | The AcrIIA15 dimer binds to promoter DNA and the SaCas9-sgRNA complex simultaneously. a** Crystal structure of full-length AcrIIA15. **b,** Structural superposition of the two protomers of the AcrIIA15 dimer aligned by their CTD. **c,** Zoom-in of the polar interactions mediating the dimerization of the AcrIIA15$^{NTD}$. **d** Overlay of SEC profiles characterizing the dimerization ability of wild-type AcrIIA15 (full-length) or key point mutants. **e** EMSA charactering mutations in key residues of AcrIIA15 that mediate dimerization by mixing Cy3-labeled IR1 dsDNA with mutants at a molar ratio of 1:5. EMSA samples were separated on a 5% native gel and visualized using a FluorChem system. **f** Overall structure of the SaCas9-sgRNA-AcrIIA15-IR1 quaternary complex. Source data are provided as a Source Data file.

the CTDs further contribute to the stability of the dimer (Supplementary Fig. 10d). Overall, the HTH domain plays a crucial role in the stabilization of the dimer, which is further reinforced by interactions between the C-terminal domains.

We generated three mutants of AcrIIA15, namely R2A, L44A, and R2A/L44A, to investigate their impact on dimerization and DNA binding. These mutants fold properly as measured by CD spectroscopy (Supplementary Fig. 4d). Through SEC analysis, we found that the R2A mutant of AcrIIA15 retained its dimeric state, similar to the wild-type protein. However, the L44A and R2A/L44A mutations resulted in monomeric forms (Fig. 6d). Furthermore, EMSA results demonstrated that the R2A mutant exhibited a mild decrease in DNA-binding ability compared to the wildtype, while the L44A and R2A/L44A mutants completely abolished DNA binding (Fig. 6e). These findings suggest that dimerization is essential for the recognition of dsDNA by AcrIIA15, consistent with the structural observation that the HTH domains of a dimer bind to the inverted-repeat sequence of the DNA. Our results align well with a previous study demonstrating that the HTH dimer of AcrIF24 interacts with DNA[29].

### The AcrIIA15-DNA complex tethers two SaCas9-sgRNA effectors together

The NTD of an AcrIIA15 dimer binds to the IR DNA, while the CTD can bind to the SaCas9-sgRNA complex as a monomer. This raises the question of whether AcrIIA15 can bind both the promoter DNA and Cas9 simultaneously. To address this, we reconstituted the SaCas9-sgRNA-AcrIIA15-IR quaternary complex by incubating these four

components together, followed by SEC (Supplementary Fig. 11a). We then determined its cryo-EM structure at a resolution of 3.82 Å (Fig. 6f). Although the resolution of this structure is not high, it is sufficient to reveal the relative locations of all subunits.

In this quaternary complex, one AcrIIA15 dimer binds one IR DNA via its HTH domains, while the two CTD domains of the dimer bind two different SaCas9-sgRNA complexes. The interactions between IR1 and AcrIIA15$^{NTD}$, and between SaCas9-sgRNA and AcrIIA15$^{CTD}$ in this complex are the same as the ones that we observed in their individual structures. However, a structural comparison of this quaternary complex with the AcrIIA15-DNA complex revealed that the position of the AcrIIA15 CTD relative to its NTD is shifted from its location in the dimer:IR1 structure. Although the DNA and HTH domains were well-aligned, both CTD domains are rotated away to accommodate SaCas9 in the quaternary complex (Supplementary Fig. 11b). This indicates that the AcrIIA15 dimer undergoes a significant conformational change upon binding to Cas9. Additionally, the short loop connecting the NTD and CTD of AcrIIA15 is flexible, enabling the structural rearrangement of AcrIIA15 in distinct states. Due to the presence of two inverted repeats located upstream of the promoter region, we posit that two AcrIIA15 dimers together with four SaCas9-sgRNA effectors could simultaneously bind to the AcrIIA15 promoter region. Similar to other HTH-containing transcriptional repressors and previously studied Aca proteins, our findings suggest that the binding of AcrIIA15 obstructs RNA polymerase from accessing the promoter, effectively inhibiting the transcription of the *acrIIA15* gene.

## Discussion

Our structural and biochemical studies demonstrate that AcrIIA15 functions as a dimer to simultaneously inhibit both anti-CRISPR operon transcription and SaCas9-sgRNA DNA recognition. The NTD domains of two AcrIIA15 molecules bind to inverted repeat DNA sequences upstream of the Acr operon in a sequence-specific manner, while the CTD interacts with the SaCas9-sgRNA complex.

### AcrIIA15 does not affect sgRNA loading to SaCas9

Our studies revealed that the CTD of AcrIIA15 binds in the crevice between the PI and WED domains of SaCas9, competing for the dsDNA binding site, but does not compete with sgRNA for binding to SaCas9 as a previous study reported[27]. In addition to our in vitro cleavage assays demonstrating that the order of addition of AcrIIA15 or sgRNA to Cas9 was not important for ternary complex formation (Supplementary Fig. 2), we also confirmed this finding structurally, showing that AcrIIA15 forms direct interactions with the first nucleotide of the repeat of sgRNA and that both coexist in the same stable complex (Fig. 3d). We speculate that the discrepancy between our results and the previous study may be due to the relatively low salt concentration used in the original paper's EMSA experiments. As full-length AcrIIA15 is not stable in low salt buffer, we believe that the SaCas9-AcrIIA15 complex may have precipitated before binding to sgRNA, leading to the absence of an RNA shift in the EMSA.

### Combination of transcriptional repression and cleavage inhibition in one anti-CRISPR protein

*acr* and *aca* genes are commonly co-encoded within operons, implying their collaborative functionality. During early stages of phage infection, prompt expression of Acr(s) safeguards the integrity of the phage genome[23]. AcrIIA15 employs its CTD to mimic the target DNA's PAM sequence, effectively blocking target DNA binding. This strategy is observed in other type II Acrs such as AcrIIA2, AcrIIA4, and AcrIIC5 proteins. However, excessive Acr transcription becomes detrimental to phages, disrupting the expression of crucial phage structural genes at later stages[23]. To combat this problem, Aca proteins, frequently found adjacent to Acr genes in the operon, possess a HTH motif to specifically bind to the Acr-Aca promoter region and repress Acr gene transcription, acting as a regulatory brake on Acr expression to benefit phages. AcrIIA15, by combining the HTH domain and Cas9 inhibitory domain within a single protein, exhibits dual functionality as a bi-functional anti-CRISPR protein capable of both transcriptional repression and CRISPR-based cleavage inhibition. We speculate that the fusion of Aca and Acr likely emerged from an accidental evolutionary event.

A previous study showed that AcrIIA1 can sense the level of Cas9, and overexpression of Cas9 will result in an increase of AcrIIA1 expression[24]. This implies that Cas9 sensing can somehow de-repress the transcription of the Acr operon. Due to the similarity between AcrIIA1 and AcrIIA15, we speculate that AcrIIA15 may also be able to sense the level of Cas9 and potentially transmit that signal directly to the Acr promoter region in vivo.

The HTH domains for all known fused type II anti-CRISPRs are located at the NTD, whereas type I-F anti-CRISPR AcrIF24 has a C-terminal HTH domain[29]. Both AcrIIA15 and AcrIF24 can form dimers and bind to the Acr promoter DNA to repress its transcription. Mutants disrupting the dimerization will abolish the binding between these Acrs and the inverted-repeat regions, thereby relieving the transcriptional repression. Despite these similarities, interestingly, the HTH domain of AcrIF24 alone cannot bind to the promoter DNA, in stark contrast to AcrIIA15 in which the NTD can function as a transcriptional repressor alone.

In conclusion, our study reveals that AcrIIA15 protein employs its CTD to inhibit SaCas9 cleavage activity while its HTH motif acts as a transcriptional repressor to regulate *acr* gene expression. This bi-functional nature of AcrIIA15 highlights the adaptive strategies employed by phages to evade CRISPR-Cas immunity systems. Overall, our findings enhance our understanding of the intricate mechanisms employed by phages to modulate CRISPR-Cas systems, providing insights into the arms race between phages and their bacterial hosts.

## Methods

### Protein expression and purification

The ORF of SaCas9 (encoding residues 1-1053) was cloned into expression vector pET28a-Sumo with a His6-Sumo tag at the N terminus. The ORFs of AcrIIA15 encoding full-length (residues 1-170), CTD (residues 57-170), and NTD (residues 1-62) portions of AcrIIA15 were cloned into an expression vector pET30b with a His6-tag at the C terminus or into pET28a-Sumo with a His6-Sumo tag at the N terminus. All AcrIIA15 mutants were prepared from the pET30b vector-derived constructs using site-directed mutagenesis.

SaCas9 proteins were overexpressed in *E. coli* Rosetta (DE3) (Novagen), and all AcrIIA15 proteins in *E. coli* BL21 (DE3) (Novagen). Cells were cultured at 37 °C until OD600 reached 0.8 and then induced with 0.2 mM isopropyl β-D-1-thiogalactopyranoside (IPTG) at 18 °C for 12 h. Next, cells were harvested and lysed by sonication in buffer containing 20 mM Tris-HCl pH 7.5 and 0.5 M NaCl at 4 °C. After centrifugation, the supernatant was incubated with Ni²⁺-Sepharose resin (GE Healthcare).

For SaCas9 and AcrIIA15CTD with His6-Sumo tags, the bound proteins were eluted with lysis buffer supplemented with a nonlinear increasing gradient of imidazole. Eluted SaCas9 and AcrIIA15CTD proteins were digested with His-tagged ubiquitin-like protein 1 (Ulp1) protease and dialyzed against buffer containing 20 mM Tris-HCl pH 7.5, 0.3 M NaCl at 4 °C for 2 h. Further, the flow-through collections were purified by HiTrap SP and Q HP columns (GE Healthcare), respectively. For full-length AcrIIA15 and AcrIIA15NTD with His6-Sumo tags, eluates from the first Ni column were digested by Ulp1 at room temperature and dialyzed against buffer containing 20 mM Tris-HCl pH 7.5, 0.3 M NaCl for ~12 h after the first nickel column. Next, the Ulp1-treated AcrIIA15 and AcrIIA15NTD samples were further purified by Q HP and Heparin columns, respectively, and then a second Ni column to remove the His6-Sumo tag and uncleaved fusion proteins.

For full-length or truncated AcrIIA15 with a His6-tag at the C terminus, the protein bound to Ni²⁺-Sepharose resin was eluted by buffers containing 20 mM Tris-HCl pH 7.5, 0.3 M NaCl supplemented with 100, 200 and 300 mM imidazole successively. The collected sample was loaded onto a HiTrap Q HP column and eluted with a linear gradient of 0.3–1.0 M NaCl.

### In vitro transcription and purification of sgRNA

We used in vitro transcription with T7 RNA polymerase and a linearized plasmid template to synthesize 98-nucleotide (nt) sgRNA. Transcription reactions were performed at 37 °C for 5 h in buffer containing 100 mM HEPES-KOH pH 7.9, 20 mM MgCl₂, 30 mM DTT, 2.7 mM each NTP, 2 mM spermidine, 0.1 mg/mL T7 RNA polymerase, and 40 ng/μL linearized plasmid DNA template. The sgRNA was then purified by gel electrophoresis on a 12% denaturing (8 M urea) polyacrylamide gel and an Elutrap system. Finally, the sgRNA was resuspended in DEPC (diethylpyrocarbonate) H₂O.

### DNA substrates

All short DNA substrates used for crystallization and EMSA experiments were purchased from Sangon Biotech. Each single-stranded (ss) DNA oligonucleotide was dissolved in an annealing buffer containing 20 mM Tris-HCl pH 7.5, 100 mM NaCl, and 10 mM MgCl₂. To obtain the double-stranded (ds) DNA, two ssDNA oligonucleotides were mixed at a molar ratio of 1 to 1.1 and heated at 95 °C for 10 min, and then cooled to room temperature gradually in order to anneal the strands. Nucleic acid sequences used in this study are summarized in Supplementary Table S1.

## Size-Exclusion Chromatography Assay

For the binding assay of apo-SaCas9 with AcrIIA15^CTD, we mixed purified SaCas9 and AcrIIA15^CTD at a molar ratio of 1:2. To measure binding of AcrIIA15^CTD to the SaCas9-sgRNA complex, we mixed purified SaCas9, sgRNA, and AcrIIA15 at a molar ratio of 1:1.1:2, adding each component sequentially in this order and setting a 30-min incubation in between. To assess binding of DNA-bound SaCas9-sgRNA with AcrIIA15^CTD, we first mixed SaCas9 and sgRNA in a molar ratio of 1:1.1 and then incubated on ice for 30 min. Next, we added dsDNA at a molar ratio of 1.2 and incubated the mixture for 40 min at room temperature. Finally, AcrIIA15^CTD was added at a molar ratio of 2 and another 30-min incubation period was performed on ice. To avoid potential cleavage of target dsDNA, 2 mM EDTA was added to the incubation system before addition of dsDNA. A Superdex 200 increase 10/300 GL column (GE Healthcare) was run in buffer containing 20 mM Tris pH 7.5, 300 mM NaCl.

For the binding assays of AcrIIA15 with dsDNA containing inverted repeat 1 (IR1) or its variants, AcrIIA15 and dsDNA were mixed at a molar ratio of 2 to 1.1 on ice for 30 min. A Superdex 200 increase 10/300 GL column was used in buffer containing 20 mM Tris pH 7.5, 100 mM NaCl.

To assess the stoichiometric ratio of AcrIIA15 binding to the full-length promoter IR1-IR2 dsDNA, dsDNA and AcrIIA15 were mixed at different molar ratios (1:2, 1:3, 1:4, and 1:6) on ice for 30 min. A Superdex 200 increase 10/300 GL column was used in buffer containing 20 mM Tris pH 7.5, 100 mM NaCl.

To test the dimerization state of wild-type or mutated AcrIIA15, we used a Superdex 75 column (GE Healthcare) in buffer containing 20 mM Tris-HCl pH 7.5 and 0.5 M NaCl.

Relevant fractions from each column elution were evaluated by SDS-PAGE followed by Coomassie staining, and fractions containing nucleic acids were subjected to a 20% urea gel after phenol chloroform extraction.

## In vitro cleavage assay

The target DNA sequence containing a 5′-N$_2$GAAT-3′ PAM and a 20-bp protospacer complementary to the guide region of the sgRNA was cloned into a modified pUC19 vector linearized by HindIII-HF (New England Biolabs).

For SaCas9-sgRNA-AcrIIA15/AcrIIA15^CTD complex, 100 nM SaCas9 was incubated with 110 nM sgRNA for 30-min incubation on ice, and then different concentrations of AcrIIA15 was added for another 30-min incubation on ice. The preparation of SaCas9-AcrIIA15-sgRNA and SaCas9-AcrIIA15^CTD-sgRNA complexes are the same except with different incubation order. The resulting enzymatic complexes were incubated with 200 ng target DNA in a 10 μL reaction in buffer containing 20 mM Tris-HCl pH 7.5, 300 mM KCl, 10 mM MgCl$_2$, 1 mM DTT and 5% glycerol for 15 min at 37 °C. Reactions were quenched by adding 2 μL gel-loading dye containing 10 mM EDTA (New England Biolabs). The reaction products were run on 1% agarose gels and stained with ethidium bromide for product detection.

## Electrophoretic mobility shift assays

To test the effect of AcrIIA15^CTD on the ability of SaCas9-sgRNA to bind target dsDNA, we performed electrophoretic mobility shift assays (EMSA) using a 53-bp dsDNA oligonucleotide substrate with the 5′-end of the target strand (TS) labelled with Cy3. 3 μM SaCas9-sgRNA complex was incubated in binding buffer containing 20 mM Tris pH 7.5, 200 mM NaCl on ice for 30 min, and different concentrations of AcrIIA15^CTD were added for another 30-min incubation on ice. Then, the pre-formed complexes were mixed with 0.6 μM dsDNA at room temperature for 40 min. To avoid potential cleavage of dsDNA, 10 mM EDTA was added before dsDNA. Products were separated on a 5% native polyacrylamide gel at 100 V for 40 min. DNA was visualized using a FluorChem system.

To test the binding of AcrIIA15 with promoter DNA, varying concentrations of wild-type full-length AcrIIA15 (2, 4, 10, 20 and 40 μM), AcrIIA15^NTD (2, 4 and 10 μM) and AcrIIA15^CTD (2, 4 and 10 μM) were mixed with 2 μM inverted repeat DNAs (IR1-IR2, IR1 or IR2) in binding buffer containing 20 mM Tris pH 7.5, 300 mM NaCl and incubated on ice for 30 min. The DNA-protein complexes were separated on a 5% native polyacrylamide gel at 100 V for 35 min, and stained with ethidium bromide for product detection.

To test the effect of AcrIIA15 mutations on promoter DNA binding, 0.4 μM 5′-Cy3-labeled IR1 was incubated with each of the AcrIIA15 mutants (2 μM) in buffer containing 20 mM Tris pH 7.5, 300 mM NaCl for 30 min at 4 °C. Products were separated using a 5% native polyacrylamide gel and visualized with the FluorChem system.

## Circular Dichroism Spectroscopy

The wild-type and mutated AcrIIA15 and AcrIIA15^CTD proteins were incubated in buffers containing 10 mM Na$_2$HPO4, 2 mM KH$_2$PO4, 0.5 M NaCl and 10 mM Na$_2$HPO4, 2 mM KH$_2$PO4, 0.3 M NaCl, respectively. The protein samples were scanned on a Chirascan Plus CD Spectropolarimeter from 202 to 260 nm. Each scan was performed at 4 °C in 1 nm increments.

## In vivo reporter assays

The sequences encoding wild-type or mutant AcrIIA15 proteins were cloned into the pET30b vector (Kanamycin) under control of the T7 promoter and lac operator. The *sfgfp* gene was cloned into a modified pCDFDuet vector (Streptomycin) with the promoter region of AcrIIA15 upstream. The pCDFDuet-AcrIIA15 promoter-sfGFP and pET30b-AcrIIA15 plasmids were co-transformed into the *E. coli* BL21 (DE3) strain and incubated on Streptomycin-Kanamycin double-resistant plates. A single colony was inoculated into a culture of 100 ml LB medium supplemented with 25 mg/L Kanamycin and 25 mg/L Streptomycin, and then cultured for 12 h at 37 °C.

Next, 3 mL saturated overnight cultures of *E. coli* were diluted to OD$_{600}$ = 0.8 in LB supplemented with 25 mg/L Kanamycin and 25 mg/L Streptomycin with 1 mM IPTG. After culturing at 37 °C for 5.5 h, 200 μL of each sample was transferred to a 96-well transparent plate for sfGFP fluorescence intensity measurement. A Varioskan Flash Microplate Reader was used, and the excitation and emission wavelengths were 490 nm and 509 nm, respectively. The values of fluorescence intensity were normalized to OD$_{600}$ and then plotted with Prism 7. Each experiment has three independent repeats.

## Surface plasmon resonance assays

Experiments were performed with a Biacore 8 K optical biosensor instrument using CM5 chips (GE Healthcare, Biacore) in a single-cycle format. All the proteins used in SPR were buffer-exchanged to PBS buffer containing 10 mM Na$_2$HPO4, 2 mM KH$_2$PO4, 0.3 M NaCl, which was filtered through 0.22 μm micro-membrane. AcrIIA15 was captured by the immobilized His antibodies on a CM5 sensor surface using standard amine-coupling procedures. His antibody was diluted to 50 μg/mL with coupling buffer (10 mM sodium acetate, pH 4.5), and then was injected on the surface and remaining activated groups were blocked with 1 M ethanolamine-HCl, pH 8.5. IR1 or IR2 DNA was serially diluted by PBS buffer to a gradient of concentrations (1.25, 2.5, 5, 10, 20 nM). The DNAs were then flowed over the chip surface and the response units were measured. The binding response was normalized by subtracting the value of control channel to eliminate the bulk effects. The data were fitted to a 1:1 interaction model using the kinetics evaluation software from Biacore. Values are means ± SD from three technical replicates.

## Complex reconstitution, crystallization and structure determination

Complexes of AcrIIA15 (full-length or NTD) with DNA were prepared by mixing purified AcrIIA15 with DNA at a molar ratio of 2:1.1 in buffer

containing 20 mM Tris pH 7.5, 100 mM NaCl for a 30-min incubation on ice. The absorbances at 280 nm of apo-AcrIIA15, AcrIIA15-DNA, and AcrIIA15[NTD]-DNA samples were 44, 20 and 25, respectively.

Hanging-drop vapor-diffusion method was used for crystal growth. All crystals were obtained by mixing 1 μL sample with 1 μL reservoir solution and incubated at 16 °C. Crystals of the full-length AcrIIA15 were grown from 0.2 M Trimethylamine N-oxide dihydrate, 0.1 M Tris pH 8.82, 18% w/v PEG 2,000. Crystals of the AcrIIA15-DNA complex were grown in 0.2 M Sodium chloride, 0.1 M HEPES-NaOH pH 7.5, 25% w/v PEG 3,350. Crystals of the AcrIIA15[NTD]-DNA complex were grown in 1.6 M sodium citrate pH 6.05. Crystals of apo-AcrIIA15 and AcrIIA15-DNA were cryoprotected using the corresponding reservoir solution supplemented with 20% ethylene glycol, and crystals of AcrIIA15[NTD]-DNA with 20% glycerol, and finally flash-frozen in liquid nitrogen.

All diffraction datasets were collected at beamline BL02U1 or BL19U1 at the Shanghai Synchrotron Radiation Facility (SSRF) and processed with HKL2000[34] or HKL3000[35]. All crystal structures were solved by molecular replacement (MR) using PHENIX PHASER[36]. The phasing model used for the structure of full-length AcrIIA15 was a AcrIIA15[CTD] structure which was solved previously using single-wavelength anomalous dispersion (SAD) method. The coordinates of AcrIIA15[NTD] were built manually in COOT[37]. Then, the resolved AcrIIA15 structure was used as the starting model for AcrIIA15-DNA and AcrIIA15[NTD]-DNA complexes. After the initial phases were obtained, the atomic models were manually built and adjusted in COOT, followed by refinement using PHENIX[38]. Iterative cycles of adjustment and refinement were performed before a qualified model was obtained. Data collection and structure refinement metrics are listed in Supplementary Table S2.

## Cryo-EM sample preparation and data acquisition

To assemble the SaCas9-sgRNA-AcrIIA15[CTD] ternary complex, we mixed SaCas9, AcrIIA15[CTD], and sgRNA sequentially at a molar ratio of 1:1.1:1.2. To assemble the SaCas9-sgRNA-AcrIIA15-IR quaternary complex, we mixed SaCas9, sgRNA, AcrIIA15, and IR1 DNA sequentially at a molar ratio of 1:1.1:0.9:0.6. A 30-min incubation on ice was performed after each addition before proceeding to the next component. For both complexes, the incubated samples were purified on a Superdex 200 Increase 10/300 GL column. Buffers used for preparation of the ternary and quaternary complexes are (20 mM Tris pH 7.5 and 0.2 M NaCl) and (20 mM Tris pH 7.5 and 0.1 M NaCl), respectively. A 0.25 mL aliquot collected from the peak elution fraction of SEC was used for cryo-EM sample preparation. The absorbance at 280 nm of the samples are 0.7 and 2.5 for the ternary and quaternary complexes, respectively.

C322 Cu 300 mesh grids and C411 Cu 400 mesh grids were glow discharged in a $O_2$-Ar condition for 50 s, and used for the SaCas9-sgRNA-AcrIIA15[CTD] and SaCas9-sgRNA-AcrIIA15-IR complex sample preparation, respectively. A 5 μL sample droplet was applied to the grid followed by blotting for 5.0 s at 100% humidity and 4 °C, and flash-frozen in liquid ethane using a Vitrobot Mark IV (Thermo Fisher Scientific, USA).

Grids were imaged with a 300 kV Titan Krios (Thermo Fisher Scientific, USA) equipped with a K2 Summit direct electron detector (Gatan, USA) and a GIF-Quantum energy filter. A calibrated magnification of 130,000× was used for collecting dose-fractioned super-resolution movie stacks binned to a pixel size of 1.04 Å by SerialEM[39]. For each movie stack, a total exposure time of 6.4 s was used, generating 32 movie frames with a total dose of ~60 electrons per Å². The defocus range was set to be between −1.4 and −1.8 μm for the SaCas9-sgRNA-AcrIIA15[CTD] ternary complex, and between −1.0 and −1.5 μm for the SaCas9-sgRNA-AcrIIA15-IR quaternary complex.

## Single-particle cryo-EM data processing

For the SaCas9-sgRNA-AcrIIA15[CTD] complex, a total of 2823 movie stacks were imported to RELION 3.1[40] and motion-corrected. Parameters of the contrast transfer function (CTF) were estimated using CTFFIND-4.1[41]. A set of 954 particles was picked manually to generate two-dimensional (2D) averages for subsequent template-based auto-picking. A total of 3,926,680 particles were automatically picked, and then extracted for 2D classification to exclude false and bad particles. After 3 rounds of reference-free 2D classification, 370,321 particles were selected for 3D classification. First, an initial model was obtained and used for 3D classification. Three classes out of six containing 237,732 particles were selected for 3D reconstruction. After 3D refinement, a model with 3.79 Å resolution was obtained. Afterward, CTF refinement and Bayesian polishing were implemented. Subsequently, the processed 237,732 particles were imported to cryoSPARC[42] for non-uniform refinement. Finally, a map with 3.31 Å resolution was reported according to the golden-standard Fourier shell correlation (GSFSC) criterion (Supplementary Fig. S12).

For the SaCas9-sgRNA-AcrIIA15-IR complex, a total of 3,084 movie stacks were imported to cryoSPARC. Movie stacks were aligned using patch motion correction, and Contrast transfer function (CTF) parameters were estimated using Patch CTF. A set of 1,265 particles was picked manually to generate 2D averages for subsequent template-based auto-picking. A total of 1,125,640 particles were automatically picked and screened by 2D classification. After 3 rounds of reference-free 2D classification, 594,925 particles were selected for 3D classification. One class out of three containing 139,070 particles was selected for 3D reconstruction. After non-uniform refinement, a 3.69-Å map of SaCas9-sgRNA-AcrIIA15-IR dimer was obtained. In this map, one copy of the SaCas9-sgRNA complex in the dimer shows poor electron density. To improve the local resolution, a local mask was calculated by the 'vop maximum' and 'vop minimum' program in Chimera and used for 3D classification for 139,070 particles. One class out of four containing 72,182 particles was selected for 3D reconstruction. After non-uniform refinement, a map of 3.82-Å resolution was reported according to the golden-standard Fourier shell correlation (GSFSC) criterion (Supplementary Fig. S13). The statistics of data collection and structure refinement of cryo-EM structures are listed in Supplementary Table S3. All structure figures were prepared using PyMOL[43], Chimera[44] or ChimeraX[45].

## Statistics and reproducibility

Quantification of gel bands were performed by ImageJ (1.40g), and the graphs were illustrated by GraphPad Prism (v7.04). Statistical parameters for specific experiments can be found within the relevant figure legends. All biochemical assays were repeated at least three times with robust reproducibility.

## Reporting summary

Further information on research design is available in the Nature Portfolio Reporting Summary linked to this article.

## Data availability

The atomic coordinates generated in this study have been deposited in the Protein Data Bank (PDB) under the accession code 8JFO (apo-AcrIIA15), 8JFU (AcrIIA15-DNA), 8JFR (AcrIIA15[NTD]-DNA), 8JFT (SaCas9-sgRNA-AcrIIA15[CTD]) and 8JG9 (SaCas9-sgRNA-AcrIIA15-IR). The cryo-EM maps generated in this study have been deposited in the Electron Microscopy Data Bank (EMDB) under the accession number EMD-36217 (SaCas9-sgRNA-AcrIIA15[CTD]) and EMD-36225 (SaCas9-sgRNA-AcrIIA15-IR). Two previously published atomic coordinates were used in Fig. 2: 5AXW (SaCas9-sgRNA-DNA) and 355D (a typical B-form DNA). The gels and fluorescence data generated in this study are provided in the Source Data file. Source data are provided with this paper.

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

## Acknowledgements

We thank the staff of the BL17U1 and BL19U1 beamlines at the Shanghai Synchrotron Radiation Facility for helping with crystal data collections. We would like to thank the staff of the Center for Biological Imaging, Core Facilities for Protein Science at the Institute of Biophysics, CAS for support to collect cryo-EM data. We are grateful to Yuanyuan Chen, Zhenwei Yang and Bingxue Zhou (Institute of Biophysics, CAS) for technical help with Biacore experiments. This work was supported by grants from National Key R&D Program of China (2023YFA0915000, 2023YFC3403400), the Natural Science Foundation of China (32330055, 31930065, 22121003, 31725008, 32071198, 32071444 and

91940302), the Chinese Academy of Sciences (XDB0570000 and XDB37010202), Beijing Municipal Science & Technology Commission (Z231100007223004), Beijing Natural Science Foundation (5232022), and the Youth Innovation Promotion Association of the Chinese Academy of Sciences (2021090). We are grateful to Dr. April Pawluk in Life Science Editors for help with editing the manuscript.

## Author contributions

X.D. and X.L. expressed and purified the proteins and grew crystals. J.W., X.L. and X.D. collected X-ray diffraction data, and X.D., J.W., X.L. and Z.C. solved all the crystal structures. X.D. and Z.C. carried out cryo-EM structure determination. X.D., W.S. and G.S. performed the biochemical assays. Y.W., W.S. and X.D. wrote the manuscript. Y.W. designed this project and supervised all the structural and biochemical studies.

## Competing interests

The authors declare no competing interests.
