## [Peer Review File · Nature Communications]

An anti-CRISPR that represses its own transcription while blocking Cas9-target DNA bindingREVIEWER COMMENTS

Reviewer #1 (Remarks to the Author):

Yanli Wang and coworkers report on structural and mutational studies of anti-CRISPR protein AcrIIA15 bound to *Staphylococcus aureus* Cas9 (saCas9) to gain mechanistic insights into anti-CRISPR action in this system. They demonstrate that the HTH fold of the N-terminal domain of AcrII15 adopts a dimeric alignment and targets inverted repeats of IR1-IR2 DNA, thereby inhibiting anti-CRISPR operon transcription. The C-terminal domain of Acr15 as a monomer targets SaCas9-sgRNA on the same surface used for recognition of the protospacer adjacent motif (PAM) element of dsDNA, thereby blocking PAM recognition. These recognition principles were identified through x-ray and cryo-EM analysis, with key intermolecular contacts validated from mutational studies. The authors then undertake a structural analysis demonstrating that AcrIIA15 can simultaneously target both promoter DNA and SaCas9-sgRNA, thereby nicely establishing AcrIIA15's dual functionalities in both autoregulation of transcription and Cas9 target binding.

The paper reports on high quality structural and mutational data, as well as outlines interesting mechanistic concepts with clarity. I recommend publication in its current form.

Dinshaw Patel

Reviewer #2 (Remarks to the Author):

The authors performed a structural and biochemical analysis of the anti-CRISPR-Cas9 protein AcrIIA15 that inhibits *Staphylococcus aureus* Cas9 (SaCas9). AcrIIA15 was discovered in 2020 by Watters et al. (www.pnas.org/cgi/doi/10.1073/pnas.1917668117) who reported that this anti-CRISPR-Cas9 protein 1) binds to inverted repeat (IR) sequences of its promoter region through its N-terminal HTH motifs, thereby likely acting as a transcriptional repressor, and 2) prevents sgRNA from binding to SaCas9, via its C-terminal domain, thereby inhibiting the SaCas9 DNA cleavage activity. In the submitted manuscript, the authors determined the crystal structure of AcrIIA15 - a dimeric protein - and of its N-terminal domain bound to one IR sequence, and they used alanine substitutions along with Electrophoretic Mobility Shift Assays (EMSA) to reveal the molecular determinants of sequence-specific interactions between the AcrIIA15 HTH motifs and its promoter sequence. They also showed using cryo-Electron Microscopy and EMSA (+/- alanine substitutions) that the AcrIIA15 C-terminal domain binds to the PAM-binding site of the SaCas9-sgRNA complex, thereby preventing target DNA binding. These results provide structural and biochemical evidences of the molecular mechanism used by AcrIIA15 to inhibit SaCas9,

and hence revisit the AcrIIA15 mode of inhibition presented in the 2020 paper. The authors also detected interactions between SaCas9 and AcrIIA15 in Size Exclusion Chromatography experiments, as previously observed by Watters et al. However, in their experimental conditions with a higher ionic strength, the Cas9-sgRNA-AcrIIA15 do assemble regardless of the order of incubation between AcrIIA15, SaCas9 and the sgRNA. Lastly, the authors also determined the cryoEM structure of the AcrIIA15 dimer bound to two SaCas9-sgRNA complexes via its C-terminal domains and to one IR sequence via its N-terminal HTH motifs.

Several examples of anti-CRISPR proteins containing 1) HTH motifs that binds to IR sequences of their promoter region, and 2) a separate domain involved in the inhibition of the CRISPR-Cas activity, have previously been reported (AcrIIA1: Osuna et al., <https://doi.org/10.1016/j.chom.2020.04.002> ; AcrIF24: Kim et al. <https://doi.org/10.1093/nar/gkac880> ; AcrIIA13-AcrIIA14: Watters et al., www.pnas.org/cgi/doi/10.1073/pnas.1917668117). Noteworthy, the transcriptional repression activity associated to their DNA-binding capability has been experimentally demonstrated only for AcrIIA1 and AcrIF24 using transcriptional reporter assays. There is no such experimental evidence for AcrIIA15 in the Watters et al. paper, which states “AcrIIA13–AcrIIA15 also are fusion proteins that link an anti-SaCas9 Acr with a DNA binding domain that is likely to regulate the expression of its own operon”, nor in the submitted manuscript. The added-value of this study for the anti-CRISPR field is the structural and biochemical evidences that AcrIIA15 prevents SaCas9 from binding to its target DNA, thereby updating our knowledge of its molecular mechanism. Therefore, it is the editor’s decision to determine whether this manuscript merit publication in Nature Communications with respect to the journal scope and associated standards.

Specific comments

1. In the absence of experimental evidences for transcriptional repression, like those presented in the AcrIIA1 (Osuna et al., <https://doi.org/10.1016/j.chom.2020.04.002>) and AcrIF24 (Kim et al., <https://doi.org/10.1093/nar/gkac880>) papers, several statements throughout the text, listed hereafter, should be amended. Noteworthy, the authors do mention in p.9 “our findings suggest that the binding of AcrIIA15 obstructs RNA polymerase from accessing the promoter, effectively inhibiting the transcription of the *acrIIA15* gene.”

-Title: “An anti-CRISPR that represses its own transcription ...”

-Abstract: “These findings shed light on AcrIIA15's inhibitory mechanisms and its autoregulation of transcription”

-p5: “The NTD of AcrIIA15 protein contains a conserved HTH motif, and previous work has shown that the HTH motif acts as a transcriptional repressor by binding to the promoter region of the *acrIIA15* gene itself, ...”.

There is no such evidence in the Watters et al. paper.

-p9: “Our structural and biochemical studies demonstrate that AcrIIA15 functions as a dimer to simultaneously inhibit both anti-CRISPR operon transcription and SaCas9-sgRNA DNA recognition.”

-p11: "In conclusion, our study reveals that AcrIIA15 protein employs its CTD to inhibit SaCas9 cleavage activity while its HTH motif acts as a transcriptional repressor to regulate *acr* gene expression."

2. p3: "Importantly, the order of incubation of AcrIIA15CTD with SaCas9 and the single-guide RNA (sgRNA) ..".

- This raises the question whether the situation is the same with the full-length AcrIIA15.

3. p3: "AcrIIA15CTD can bind to SaCas9 in both the apo form and sgRNA-bound state".

- How can this be explained in the light of the SaCas9-sgRNA-AcrIIA15 structure?

4. p3: "AcrIIA15CTD is situated in the cleft between the PI and WED domains of SaCas9".

- PI and WED should be defined.

5. p4: "AcrIIA15CTD is composed of five α -helices and a β -sheet consisting of three β -strands".

- It would be nice to show this structure on its own in Fig.2. Is the CTD a new fold?

6. p4: "our findings suggest that AcrIIA15CTD inhibits the DNA cleavage activity of SaCas9 by preventing it from binding to target DNA." and "suggesting that AcrIIA15CTD binding directly blocks the PAM recognition site in SaCas9".

- Isn't "show" more appropriate than "suggest"?

7. p4. "To evaluate the importance of specific residues in AcrIIA15CTD for its binding to the SaCas9-sgRNA complex, we performed in vitro DNA cleavage assays using SaCas9 and various point mutants of AcrIIA15CTD (Fig. 3g)." and p.5 "These findings highlight the crucial role of these residues in facilitating the complex formation between AcrIIA15CTD and the SaCas9-sgRNA complex, contributing to the inhibitory function of AcrIIA15".

- The DNA cleavage assay is an indirect mean of evaluating the role of AcrIIA15 residues on SaCas9 binding. In this case, the authors evaluated the importance of some residues for the AcrIIA15 anti-CRISPR activity. However, SEC experiments, like those performed with Y162A (p5), are suitable to monitor macromolecular interactions.

- Were all mutants as stable in solution as the wt AcrIIA15? The loss of inhibition could be due to a reduced amount of active AcrIIA15 if the mutation alters the protein stability over time.

- Not all mutations shown in Fig 3,g are commented in the text. For instance, S157A abolishes the Acr activity as do Y161A and Y162A.

8. p5: "We found that alanine substitutions of residues Y97 and Y141, which interact with the PI domain of SaCas9, led to a notable decrease in the inhibitory effect of AcrIIA15CTD. In contrast, mutations of residues D125 and Y130, involved in interactions with the PAM-recognizing residues of SaCas9, had a relatively minor impact on the inhibitory activity. Importantly, the mutations of AcrIIA15 residues Y161 and Y162, which interact with the sgRNA and the WED domain, resulted in the near-complete loss of inhibition by AcrIIA15CTD. These findings highlight the crucial role of these residues in facilitating the complex formation between AcrIIA15CTD and the SaCas9-sgRNA complex, contributing to the inhibitory function of AcrIIA15."

- What is the difference between "a notable decrease" and "the near-complete loss"? The gel does show mutations that affect the AcrIIA15 activity and mutations that do not affect the AcrIIA15 activity. Based on band intensities, the effects of Y97A and Y161A are similar, and D125A is stronger than the wt. This paragraph should be amended.

9. p5: "To further investigate the impact of the Y162A mutation in AcrIIA15..."

- Why the authors have chosen to focus on this residue?

-One does not understand what "To further investigate" means.

It is hard to understand the logic between this paragraph and the previous one.

10. p5: "the full-length AcrIIA15 exhibited efficient binding to IR1, IR2 and the entire IR1+IR2, even at low concentrations (Fig. 4a). The HTH-containing NTD domain alone also binds to IRs efficiently."

- "efficient" is too vague and not appropriate.

- The molar ratio, instead of the concentration, is the parameter to be taken into account here.

- The presence of high MW bands in Fig 4a as well as in extended Fig5 should be commented.

11. p7: "Mutations of these amino acids result in a significant decrease in DNA binding ability (Extended Data Fig. 7e and 7f).

- "significant" is not appropriate.

- In Fig. 7e, N34A and S14A binds to DNA as does the wt AcrIIA15.

12. p7: "Importantly, we observed that residue R33 in helix 3 forms multiple hydrogen bonds with other amino acids in the NTD, indicating its crucial role in maintaining the stability of the NTD (Fig. 5g).

Consistent with this idea, we found that substituting R33 with alanine abolished the dsDNA binding ability of AcrIIA15 (Fig. 5f), providing further evidence of the significance of this residue in stabilizing the protein-DNA complex.”.

- What is the GF profile of R33A? One would expect a negative impact on the AcrIIA15 folding.

- If R33 only interacts with amino acids of the NTD, the loss of dsDNA binding ability observed with R33A should be due to a destabilization of AcrIIA15 rather than to a loss of protein-DNA interactions.

13. p9: “In this quaternary complex, one AcrIIA15 dimer binds one IR DNA via its HTH domains, while the two CTD domains of the dimer bind two different SaCas9-sgRNA complexes.”.

- It would be nice to see an overall view of this structure in the main figures.

14. p9: “This indicates that the AcrIIA15 dimer undergoes a significant conformational change upon binding both DNA and Cas9. Additionally, the short loop connecting the NTD and CTD of AcrIIA15 is flexible, enabling the structural rearrangement of AcrIIA15 in distinct states.”.

- The first sentence suggests that DNA binding induces conformational changes, which is misleading. Indeed, the structural data presented in this manuscript indicate that the intrinsic flexibility of the AcrIIA15 CTDs in solution enables binding to DNA as well as to Cas9-RNA complexes.

15. p10: “During early stages of phage infection, prompt expression of Acr(s) safeguards the integrity of the phage genome.”

- Is this at the level of a cell population or of a unique cell?

- Is there any reference for this? According to the cooperative behavior of anti-CRISPR phages, the rapidity of protein expression should not be an important parameter to protect a phage population.

16. p10: “However, excessive Acr transcription becomes detrimental to phages, disrupting the expression of crucial phage structural genes at later stages.”.

- This should come with a reference.

17. p10: “We speculate that the fusion of Aca and Acr likely emerged from an accidental evolutionary event.”

- Is this in agreement with the fact that several examples of such anti-CRISPR proteins have been identified?

18. p13: "To measure binding of AcrIIA15CTD to the SaCas9-sgRNA complex, we mixed purified SaCas9, sgRNA, and AcrIIA15 at a molar ratio of 1:1.1:2, adding each component sequentially in this order."

- Were there incubation times in between the consecutive additions?

19. p15: "All crystals were cryoprotected using the corresponding reservoir solution supplemented with 20% glycerol or ethylene glycol...".

- Which cryoprotectant for which crystal?

20. p13, p16

- References for HKL3000, PHASER, SerialEM, CTFFIND4.

21. p13: "The phasing model used for the structure of full-length AcrIIA15 was a AcrIIA15CTD structure which was solved previously using single-wavelength anomalous dispersion (SAD) method."

- Is this structure deposited in the PDB?

22. p13: "To assemble the SaCas9-sgRNA-AcrIIA15CTD ternary complex, we mixed SaCas9, AcrIIA15CTD, and sgRNA sequentially at a molar ratio of 1:1.1:1.2."

- This is counterintuitive. Why this order?

23. p13: "C322 Cu 300 mesh grids and C411 Cu 400 mesh grids ...".

- Quantifoil or C-flat grids?

24. p16: "an initial model was obtained and used for 3D classification ...".

-How was the initial model obtained?

24. Fig1. It would have been nice to show the elution curve of AcrIIA15CTD to be able to compare it with that of the Cas9-sgRNA-DNA-AcrIIA15CTD complex.

Reviewer #3 (Remarks to the Author):

Summary

In this manuscript, Deng et al. set out to characterize the molecular mechanisms of AcrIIA15 inhibition of SaCas9. Previous studies had suggested that AcrIIA15 acts by inhibiting sgRNA binding to SaCas9. In a series of biochemical and structural analyses, the authors refute this and tease apart the role of the AcrIIA15 N-terminal domain (NTD) in dimerization and promoter DNA binding, and the role of the C-terminal domain (CTD) in mimicking the target dsDNA to bind and inhibit SaCas9. The authors convincingly show that dimeric AcrIIA15 can simultaneously bind to promoter DNA inverted repeat (IR) regions via its NTD, and two SaCas9-sgRNA complexes via its CTD. In doing so, they have nicely investigated the structural and biochemical basis for the dual functions of AcrIIA15. This is a very strong paper which makes a great contribution to our understanding of AcrII family members and an intriguing domain fusion. However, certain experiments should be conducted to further strengthen the conclusions of this study, particularly around the statistical robustness of the biochemistry presented and the biological relevance of the findings described within the submission.

Major points

1. Cleavage gels should be quantified (e.g., percent cleaved) and statistical analyses performed to assess whether the inhibition of cleavage is significant (Main Fig 1a, Fig 3g; Extended data Fig 1a-b, Fig 2). This is also relevant for binding/EMSA gels (Main Fig 4a, Fig 5e-f, Fig 6e; Extended data Fig 3, Fig 5, Fig 7e-f) which should be quantified (e.g., percent bound) and statistical analyses performed to assess whether differences in binding are significant. Including statements of how many times these experiments were performed in each figure legend is also needed, with 3 independent experiments expected.

2. Determining the affinities of the various interactions explored in this paper would strengthen this study (e.g., AcrIIA15 for SaCas9 versus target DNA for SaCas9; AcrIIA15 for IR1 and/or IR2). As a particular example, the authors suggest that the binding affinity between AcrIIA15 and SaCas9-sgRNA is lower when the WED domain interacting residue Y162 is mutated (Extended data Fig 4). This conclusion would be substantiated by quantifying and comparing the binding affinities of WT and mutant AcrIIA15 for SaCas9-sgRNA. Understanding the affinity of these interactions might also give additional insight into how AcrIIA15 functions. E.g., is the affinity of AcrIIA15 for the IR promoter DNA higher than the affinity between AcrIIA15 and SaCas9, or vice versa?

3. In the context of AcrIIA15 activity during phage infection, AcrIIA15 would encounter a much larger stretch of dsDNA (indeed, an entire phage genome in some cases) where the Acr promoter region contains two IRs (IR1 and IR2). The authors speculate that both IRs could be simultaneously bound by two AcrIIA15 dimer:SaCas9 complexes. To strengthen this claim and increase the biological relevance of their study, the authors should attempt experiments to assess the stoichiometry of binding to the full-length promoter (IR1 + IR2) dsDNA sequence, at least by SEC.

4. For all experiments involving point mutants of the AcrIIA15 protein (e.g. Main Fig 3g, Fig 5f, Fig 6e; Extended Data Fig 7e-f), performing analyses (such as DSF, CD, etc) to determine whether the mutant proteins retain a similar structure/fold to WT AcrIIA15 protein would strengthen the conclusion that these residues are important and the lack of inhibition/DNA binding is specific to the mutation and not due to misfolding.

5. What is the functional consequence of the binding of both the IR in phage DNA and SaCas9? How does binding to one domain of AcrIIA15 affect subsequent binding to the other domain, and whether there is any allostery involved (e.g. does binding to SaCas9-sgRNA and conformational change affect the affinity of promoter DNA binding or vice versa). Does binding to the SaCas9-sgRNA complex affect transcriptional regulation of the Acr operon? The authors point to AcrIIA1 in the discussion as an example of a link between Cas9 abundance and the Acr operon. Could the authors test this experimentally in vivo using a reporter model that encodes GFP and include the constructs with and without NTD/CTD or SaCas9 to explore this?

Minor points

1. Figure 1: Would benefit from a schematic of the full-length vs CTD AcrIIA15 constructs to orient the reader (like what is shown in Fig 2a).

2. Figure 1a: Define FL as full length (FL) in the figure legend for clarity.

3. Figure 1: Inhibition of cleavage starts at a ratio of 1:10, with a ratio of 1:20-1:50 needed for a what appears to be a complete shift to uncleaved. This would suggest that the affinity is relatively low. It is not clear from the method what the exact concentrations are in addition to the molar ratios indicated.

4. Figure 1b: The authors should indicate the ratio and/or concentration of each component used in the complexing SEC experiments. E.g., was it 1:1 ratio? Based on the Coomassie gel, there appears to be a huge excess of SaCas9 compared to AcrIIA15-CTD in the major peak – was unbound CTD found in some of the later peaks?

5. The authors should comment on the observed specificity of AcrIIA15 for inhibition of SaCas9 – are there any clues as to how this specificity is achieved, given all three Cas9 orthologs bind dsDNA and would presumably be inhibited by DNA-mimicking AcrIIA15? Or does this reflect differences in the PAM motif recognised by the different orthologs? Are the AcrIIA15-interacting residues conserved amongst Cas9 orthologs? A small multiple sequence alignment of Cas9 orthologs could answer this question and be an interesting insight into AcrIIA15 specificity. This could be brought up in the discussion.

6. Figure 2: minor suggestion – clarity might be improved if the Cas9 domains were coloured in blue/green/cool tones, and the ACR domains in warm/pink tones or maybe even red to match the DNA (the pink Cas9 HNH domain is close in colour to the ACR NTD). Likewise, might be good to colour the B-form DNA in red in 2e. It would be good to define the acronyms used for different lobes and domains, if not in the figure legend then somewhere in the text particularly for a non-expert reader.

7. Figure 3: 3g: Would be good to have some indication of ratio/concentration used here. Was it 1:50 to achieve max WT inhibition? And was the ratio/concentration kept consistent for each mutant – important to state this.

8. Figure 4: 4a: At lower ratios DNA:Acr (e.g. 1:1 and 1:2), it seems that the IR1 is bound first (smaller band appears first) before the IR2 is bound (larger band appears from 1:2 onward). Why are the Acr-bound IR1 and IR2 bands two different sizes when the input DNA appears to be the same size (single band in the DNA only lane)? Was the ratio/concentration of IRI1 and IRI2 the same in the mixed experiment (4a left most)?

9. Do the IR regions (similar to IR1 and IR2) occur elsewhere in the phage genome? Are other genes possibly regulated by AcrIIA15 binding?

10. Figure 5: 5f: it was unclear from the figure annotations and figure legend whether these point mutations were introduced into the full length AcrIIA15 protein or in the AcrIIA15-NTD protein.

11. Extended data Fig 5: When assessing ability of AcrII15-NTD to bind IR1:IR2 by EMSA, we see a very different band shift pattern compared to full length AcrII15, in particular multiple 'bound' bands which increase in shift as amount of AcrII15-NTD is increased. Could the authors comment on this?

12. Extended data Fig 6B – The authors included extra bp outside of the IR1 inverted repeats in order for DNA:AcrII complex to be achieved. Does this imply that the bp outside the IR are also critical for DNA binding? And why was the decision made to have 1 bp overhangs in DNA2, why not dsDNA?

Manuscript NCOMMS-23-36510-T

'An anti-CRISPR that represses its own transcription while blocking Cas9-target DNA binding'

We thank the reviewers for their encouraging comments and critical feedback. Their suggestions are very helpful. Our point-by-point responses (in blue) to the reviewers' comments (in black) are as below:

Reviewer #1 (Remarks to the Author):

The paper reports on high quality structural and mutational data, as well as outlines interesting mechanistic concepts with clarity. I recommend publication in its current form.

We thank the reviewer for the supportive comments.

Reviewer #2 (Remarks to the Author):

The authors performed a structural and biochemical analysis of the anti-CRISPR-Cas9 protein AcrIIA15 that inhibits *Staphylococcus aureus* Cas9 (SaCas9). AcrIIA15 was discovered in 2020 by Watters et al. (www.pnas.org/cgi/doi/10.1073/pnas.1917668117) who reported that this anti-CRISPR-Cas9 protein 1) binds to inverted repeat (IR) sequences of its promoter region through its N-terminal HTH motifs, thereby likely acting as a transcriptional repressor, and 2) prevents sgRNA from binding to SaCas9, via its C-terminal domain, thereby inhibiting the SaCas9 DNA cleavage activity. In the submitted manuscript, the authors determined the crystal structure of AcrIIA15 - a dimeric protein - and of its N-terminal domain bound to one IR sequence, and they used alanine substitutions along with Electrophoretic Mobility Shift Assays (EMSA) to reveal the molecular determinants of sequence-specific interactions between the AcrIIA15 HTH motifs and its promoter sequence. They also showed using cryo-Electron Microscopy and EMSA (+/- alanine substitutions) that the AcrIIA15 C-terminal domain binds to the PAM-binding site of the SaCas9-sgRNA complex, thereby preventing target DNA binding. These results provide structural and biochemical evidences of the molecular mechanism used by AcrIIA15 to inhibit SaCas9, and hence revisit the AcrIIA15 mode of inhibition presented in the 2020 paper. The authors also detected interactions between SaCas9 and AcrIIA15 in Size Exclusion Chromatography experiments, as previously observed by Watters et al. However, in their experimental conditions with a higher ionic strength, the Cas9-sgRNA-AcrIIA15 do assemble regardless of the order of incubation between AcrIIA15, SaCas9 and the sgRNA. Lastly, the authors also determined the cryoEM structure of the AcrIIA15 dimer bound to two SaCas9-sgRNA complexes via its C-terminal domains and to one IR sequence via its N-terminal HTH motifs.

Several examples of anti-CRISPR proteins containing 1) HTH motifs that binds to IR sequences of their promoter region, and 2) a separate domain involved in the inhibition of the CRISPR-Cas activity, have previously been reported (AcrIIA1: Osuna et al., <https://doi.org/10.1016/j.chom.2020.04.002> ; AcrIF24: Kim et al. <https://doi.org/10.1093/nar/gkac880> ; AcrIIA13-AcrIIA14: Watters et al., www.pnas.org/cgi/doi/10.1073/pnas.1917668117). Noteworthy, the transcriptional repression

activity associated to their DNA-binding capability has been experimentally demonstrated only for AcrIIA1 and AcrIF24 using transcriptional reporter assays. There is no such experimental evidence for AcrIIA15 in the Watters et al. paper, which states “AcrIIA13 – AcrIIA15 also are fusion proteins that link an anti-SauCas9 Acr with a DNA binding domain that is likely to regulate the expression of its own operon” , nor in the submitted manuscript. The added-value of this study for the anti-CRISPR field is the structural and biochemical evidences that AcrIIA15 prevents SaCas9 from binding to its target DNA, thereby updating our knowledge of its molecular mechanism. Therefore, it is the editor’ s decision to determine whether this manuscript merit publication in Nature Communications with respect to the journal scope and associated standards.

We thank the reviewer for the detailed summary of the study of AcrIIA15 and critical comments. In this revision, we have provided the evidences of in vivo transcriptional repression by AcrIIA15 using Superfolder GFP protein as a reporter.

Specific comments

1. In the absence of experimental evidences for transcriptional repression, like those presented in the AcrIIA1 (Osuna et al., <https://doi.org/10.1016/j.chom.2020.04.002>) and AcrIF24 (Kim et al., <https://doi.org/10.1093/nar/gkac880>) papers, several statements throughout the text, listed hereafter, should be amended. Noteworthy, the authors do mention in p.9 “our findings suggest that the binding of AcrIIA15 obstructs RNA polymerase from accessing the promoter, effectively inhibiting the transcription of the *acrIIA15* gene.”

-Title: “An anti-CRISPR that represses its own transcription …”

-Abstract: “These findings shed light on AcrIIA15's inhibitory mechanisms and its autoregulation of transcription”

-p5: “The NTD of AcrIIA15 protein contains a conserved HTH motif, and previous work has shown that the HTH motif acts as a transcriptional repressor by binding to the promoter region of the *acrIIA15* gene itself, …” .

There is no such evidence in the Watters et al. paper.

-p9: “Our structural and biochemical studies demonstrate that AcrIIA15 functions as a dimer to simultaneously inhibit both anti-CRISPR operon transcription and SaCas9-sgRNA DNA recognition.”

-p11: “In conclusion, our study reveals that AcrIIA15 protein employs its CTD to inhibit SaCas9 cleavage activity while its HTH motif acts as a transcriptional repressor to regulate *acr* gene expression.”

We thank the reviewer for this suggestion. In our revision, we assessed AcrIIA15's transcriptional repression activity using a GFP reporter assay. We cloned the promoter region of AcrIIA15, which contains two inverted repeats, upstream of a superfolder *gfp* (*sfGFP*) reporter gene to illustrate the protein's transcriptional suppression capability. Subsequently, we quantified GFP fluorescence intensity as a measure of transcription levels.

The functionality of the AcrIIA15 promoter is evidenced by sfGFP expression in its presence (promoter + empty vector), as opposed to its absence (no Acr promoter). Upon AcrIIA15 expression (promoter + Acr WT), sfGFP expression decreased to levels comparable to the ‘no Acr promoter’

control. Notably, the AcrIIA15 K31A mutant (promoter + Acr K31A) substantially restored sfGFP expression, indicating a relief of transcriptional repression, consistent with our in vitro binding assay.

These results underscore AcrIIA15's role as an in vivo transcription repressor, and its binding to inverted repeat sequences is essential for transcriptional repression. We have incorporated this data into the manuscript as Fig. 5g.

2. p3: “Importantly, the order of incubation of AcrIIA15CTD with SaCas9 and the single-guide RNA (sgRNA) ..” .

- This raises the question whether the situation is the same with the full-length AcrIIA15.

The situation for the full-length AcrIIA15 is the same as AcrIIA15CTD. Inhibition remains unaffected by the order of AcrIIA15 and sgRNA incubation with SaCas9. The updated data has been included in Supplementary Fig. 2.

3. p3: “AcrIIA15CTD can bind to SaCas9 in both the apo form and sgRNA-bound state” .

- How can this be explained in the light of the SaCas9-sgRNA-AcrIIA15 structure?

AcrIIA15CTD establishes extensive interactions with the PAM-interacting (PI) and wedge (WED) domains, as well as the crevice formed by bridge helix (BH), phosphate lock loop (PLL) and the repeat region of sgRNA. In the presence of sgRNA, the crevice is formed by BH, PLL and the sgRNA, allowing AcrIIA15CTD to bind to the SaCas9-sgRNA binary complex with high affinity. In the absence of sgRNA, the crevice cannot form, and only PI and WED domains can mediate the interactions between apo-SaCas9 and AcrIIA15CTD, resulting in reduced binding affinity between apo-SaCas9 and AcrIIA15CTD. As a result, AcrIIA15CTD can bind to both apo- and sgRNA-bound SaCas9, albeit with significantly different binding affinities.

4. p3: “AcrIIA15CTD is situated in the cleft between the PI and WED domains of SaCas9.” .

- PI and WED should be defined.

Thanks for the suggestion. The PAM-interacting (PI) domain and wedge (WED) domain are two crucial domains responsible for PAM recognition. We have included the definitions of the PI and WED domains in this revision.

5. p4: “AcrIIA15CTD is composed of five α -helices and a β -sheet consisting of three β -strands.” .

- It would be nice to show this structure on its own in Fig.2. Is the CTD a new fold?

Thanks for the suggestion. The structure of AcrIIA15CTD alone from SaCas9-sgRNA-AcrIIA15CTD ternary complex has been displayed in Fig. 2e. DALI server search shows no known structure of similar folds to AcrIIA15CTD, so the CTD is a new fold. This description has been added in the revised manuscript.

6. p4: “our findings suggest that AcrIIA15CTD inhibits the DNA cleavage activity of SaCas9 by preventing it from binding to target DNA.” and “suggesting that AcrIIA15CTD binding directly

blocks the PAM recognition site in SaCas9.” .

- Isn't "show" more appropriate than "suggest" ?

We thank the reviewer for this suggestion. The word "suggest" has been replaced with "show" in these two sentences.

7. p4. "To evaluate the importance of specific residues in AcrIIA15CTD for its binding to the SaCas9-sgRNA complex, we performed in vitro DNA cleavage assays using SaCas9 and various point mutants of AcrIIA15CTD (Fig. 3g)." and p.5 "These findings highlight the crucial role of these residues in facilitating the complex formation between AcrIIA15CTD and the SaCas9-sgRNA complex, contributing to the inhibitory function of AcrIIA15." .

- The DNA cleavage assay is an indirect mean of evaluating the role of AcrIIA15 residues on SaCas9 binding. In this case, the authors evaluated the importance of some residues for the AcrIIA15 anti-CRISPR activity. However, SEC experiments, like those performed with Y162A (p5), are suitable to monitor macromolecular interactions.

We appreciate the reviewer raising this concern and have made revision to this section.

In this part, the DNA cleavage assay was employed to identify the critical residues of AcrIIA15 that are essential for its inhibitory activity. Furthermore, we used SEC experiments to demonstrate that the decrease in inhibition potency resulted from a reduction in the binding affinity between the SaCas9-sgRNA surveillance complex and AcrIIA15 mutants. The related two paragraphs have been rewritten to provide a clearer explanation of the objectives of both the DNA cleavage and the SEC assays.

- Were all mutants as stable in solution as the wt AcrIIA15? The loss of inhibition could be due to a reduced amount of active AcrIIA15 if the mutation alters the protein stability over time.

We appreciate the reviewer's concern. All mutants are stable as the wt AcrIIA15, since every altered residue is located on AcrIIA15's surface, the hydrophobic core of the protein remains unaltered. Moreover, these mutants perform well in solution during purification. Additionally, the stability of seven mutants (D72A, Y97A, Y141A, S157A, S158, Y161A, and Y162A) with reduced cleavage inhibition potency was assessed using circular dichroism. All seven AcrIIA15 mutants exhibit spectra identical to the wild-type protein as shown in Supplementary Fig. 4, suggesting that these mutations have not induced any protein misfolding issues.

- Not all mutations shown in Fig 3,g are commented in the text. For instance, S157A abolishes the Acr activity as do Y161A and Y162A.

We thank the reviewer for this reminder. The effects of alanine substitutions of all the residues shown in Fig. 3g have been described in this revision.

8. p5: "We found that alanine substitutions of residues Y97 and Y141, which interact with the PI domain of SaCas9, led to a notable decrease in the inhibitory effect of AcrIIA15CTD. In contrast,

mutations of residues D125 and Y130, involved in interactions with the PAM-recognizing residues of SaCas9, had a relatively minor impact on the inhibitory activity. Importantly, the mutations of AcrIIA15 residues Y161 and Y162, which interact with the sgRNA and the WED domain, resulted in the near-complete loss of inhibition by AcrIIA15CTD. These findings highlight the crucial role of these residues in facilitating the complex formation between AcrIIA15CTD and the SaCas9-sgRNA complex, contributing to the inhibitory function of AcrIIA15.” .

- What is the difference between “a notable decrease” and “the near-complete loss” ? The gel does show mutations that affect the AcrIIA15 activity and mutations that do not affect the AcrIIA15 activity. Based on band intensities, the effects of Y97A and Y161A are similar, and D125A is stronger than the wt. This paragraph should be amended.

We appreciate the feedback from this reviewer. In the revision, we carried out additional cleavage assays to quantify the cleavage bands and performed the recommended statistical analyses as suggested by reviewer 3. Given that AcrIIA15CTD is more stable at higher salt concentrations, we adjusted the salt concentration of potassium chloride in the DNA cleavage assay buffer from 100 mM to 300 mM. This adjustment resulted in a slight increase in the inhibitory activity of both wild-type AcrIIA15 and all AcrIIA15 mutants. However, the relative differences between them remained unchanged. We have rewritten this paragraph accordingly.

9. p5: “To further investigate the impact of the Y162A mutation in AcrIIA15…” .

- Why the authors have chosen to focus on this residue?

We thank the Referee for raising this point. Alanine substitutions for AcrIIA15 residues Y161 and Y162 completely abolished AcrIIA15CTD's inhibitory activity, demonstrating a more pronounced effect than other mutations. Consequently, we selected Y162A as a representative to investigate why it plays a crucial role in DNA binding perturbation. In this revision, we have also included data for the Y161A mutation (Supplementary Fig. 6a). We updated this paragraph to make it clearer.

-One does not understand what “To further investigate” means.

It is hard to understand the logic between this paragraph and the previous one.

We rewrote this paragraph to make it clearer.

10. p5: “the full-length AcrIIA15 exhibited efficient binding to IR1, IR2 and the entire IR1+IR2, even at low concentrations (Fig. 4a). The HTH-containing NTD domain alone also binds to IRs efficiently.” .

- “efficient” is too vague and not appropriate.

We thank the Referee for raising this point. We conducted surface plasmon resonance (SPR) experiments to evaluate the binding affinity between the full-length AcrIIA15 and IR1, IR2. Our results demonstrated that AcrIIA15 binds to IR1 and IR2 with binding affinities at a nano-molar level, with values 7.49 ± 0.60 nM and 1.80 ± 0.66 nM, respectively (Supplementary Fig. 7a).

In the revision, we replaced this sentence by “the full-length AcrIIA15 exhibits high binding affinity

to IR1, IR2 and the entire IR1-IR2 with high affinity, at a nanomolar level”

- The molar ratio, instead of the concentration, is the parameter to be taken into account here.

We have revised this sentence.

- The presence of high MW bands in Fig 4a as well as in extended Fig5 should be commented.

In the revision, we have labeled the AcrIIA15-DNA complex and free DNA in these figures.

One IR1-IR2 DNA may bind to one or two AcrIIA15 dimers, which would result in two high MW bands seen in Fig. 4a. In extended Fig. 5 (Supplementary Fig. 7c in the revised manuscript), the situation is different, and high MW bands at the 1:10 and 1:20 molar ratios are smear. We speculate that at high doses, AcrIIA15NTD is not stable. To prevent misleading, we have removed the results at 1:10 and 1:20 molar ratios in Supplementary Fig. 7c.

11. p7: “Mutations of these amino acids result in a significant decrease in DNA binding ability (Extended Data Fig. 7e and 7f).

- “significant” is not appropriate.

- In Fig. 7e, N34A and S14A binds to DNA as does the wt AcrIIA15.

In the revision, we replaced this sentence by “Mutating N16 and S30 modestly reduces their DNA binding ability, while S14 and N34 mutations cause a slight decrease (Supplementary Fig. 9d)”.

12. p7: “Importantly, we observed that residue R33 in helix 3 forms multiple hydrogen bonds with other amino acids in the NTD, indicating its crucial role in maintaining the stability of the NTD (Fig. 5g). Consistent with this idea, we found that substituting R33 with alanine abolished the dsDNA binding ability of AcrIIA15 (Fig. 5f), providing further evidence of the significance of this residue in stabilizing the protein-DNA complex.”

- What is the GF profile of R33A? One would expect a negative impact on the AcrIIA15 folding.

We thank the Referee for raising this point. The residue R33 in helix 3 forms multiple hydrogen bonds with other amino acids in the NTD, indicating the critical role of R33 in the stability of the NTD domain (Fig. R1a). Our EMSA assay showed that the R33A mutant abolished its binding to IR1 DNA (Fig. R1b). Furthermore, we conducted the gel filtration experiments and circular dichroism spectroscopy with R33A to assess its significance for NTD folding. The results indicated that the gel filtration profile (Fig. R1c) and circular dichroism spectrum (Fig. R1d) of R33A differ from those of the WT AcrIIA15, confirming its essential role in proper NTD folding.

Fig. R1 | Characterization of residue R33 of AcrIIA15.

- Contacts between residue R33 and surrounding residues in AcrIIA15NTD.
- EMSA characterizing the DNA-binding ability of mutations in R33 and other key residues of AcrIIA15. EMSA samples were separated by 5% native gels and cy3-labeled DNA was visualized by a FluorChem system.
- SEC profiles of wildtype AcrIIA15 and R33A mutant.
- Circular dichroism spectroscopy scans of wild-type AcrIIA15 and R33A mutant.

- If R33 only interacts with amino acids of the NTD, the loss of dsDNA binding ability observed with R33A should be due to a destabilization of AcrIIA15 rather than to a loss of protein-DNA interactions.

We agree with the comment of the reviewer. In the revision, we conducted gel filtration experiments and circular dichroism spectroscopy with R33A to evaluate its importance for the stability of the NTD domain. The results clearly demonstrated significant differences between R33A and WT (Fig. R1), confirming its crucial role in proper NTD folding. Since R33 does not directly interact with DNA, we have removed the paragraph regarding the R33A mutation for the sake of clarity.

13. p9: “In this quaternary complex, one AcrIIA15 dimer binds one IR DNA via its HTH domains, while the two CTD domains of the dimer bind two different SaCas9-sgRNA complexes.”

- It would be nice to see an overall view of this structure in the main figures.

We thank the reviewer for this suggestion. The overall view of the quaternary complex is referred to Fig. 6f.

14. p9: “This indicates that the AcrIIA15 dimer undergoes a significant conformational change upon binding both DNA and Cas9. Additionally, the short loop connecting the NTD and CTD of AcrIIA15 is flexible, enabling the structural rearrangement of AcrIIA15 in distinct states.” .

- The first sentence suggests that DNA binding induces conformational changes, which is misleading. Indeed, the structural data presented in this manuscript indicate that the intrinsic flexibility of the AcrIIA15 CTDs in solution enables binding to DNA as well as to Cas9-RNA complexes.

We thank the reviewer for raising this concern.

The structural comparison between the AcrIIA15 dimer in the apo state and the AcrIIA15-IR binary complex showed that only the binding of IR1 DNA does not induce a conformational change in AcrIIA15. It's worth noting that in the apo state, the two CTD domains within a dimer are initially positioned close to each other, and the linker between the NTD and CTD domains of AcrIIA15 is highly flexible. Additionally, each CTD domain binds one Cas9-sgRNA complex. To accommodate binding to Cas9, these two CTD domains move apart upon binding to Cas9. Therefore, the binding of Cas9 induces a conformational change in AcrIIA15, resulting in a distinct relative positioning of the NTD and CTD domains in the SaCas9-sgRNA-AcrIIA15-IR quaternary complex compared to the apo AcrIIA15.

We change the first sentence to “This indicates that the AcrIIA15 dimer undergoes a significant conformational change upon binding to Cas9” in the revised manuscript.

15. p10: “During early stages of phage infection, prompt expression of Acr(s) safeguards the integrity of the phage genome.”

- Is this at the level of a cell population or of a unique cell?

The prompt expression of Acrs safeguards the integrity of the phage genome at the level of a unique cell.

- Is there any reference for this? According to the cooperative behavior of anti-CRISPR phages, the rapidity of protein expression should not be an important parameter to protect a phage population.

In the case of AcrIF1, the robust transcription of Acr at the onset of phage infection was detected through RNA extraction and qRT-PCR. This is referred to Stanley et al., *Cell*, 2019.

16. p10: “However, excessive Acr transcription becomes detrimental to phages, disrupting the expression of crucial phage structural genes at later stages.”

- This should come with a reference.

Add the reference Stanley et al., *Cell*, 2019.

17. p10: “We speculate that the fusion of Aca and Acr likely emerged from an accidental evolutionary event.”

- Is this in agreement with the fact that several examples of such anti-CRISPR proteins have been identified?

This is just our speculation. With the exception of AcrIF24, all known instances of bifunctional anti-CRISPR proteins originate from type II CRISPR-Cas systems, and the Aca proteins are located at the N-terminus of anti-CRISPR proteins, so these type II fusion Acrs could be classified into one category. Based on the observation of our SaCas9-sgRNA- AcrIIA15-IR1 quaternary structure, the fusion of Aca to the C-terminus of Acr will not favor the formation of quaternary complex, and the fusion protein will not be bi-functional. Thus, it's possible that both unintentional gene transfers and evolutionary selection led to the Aca-Acr fusion.

18. p13: “To measure binding of AcrIIA15CTD to the SaCas9-sgRNA complex, we mixed purified SaCas9, sgRNA, and AcrIIA15 at a molar ratio of 1:1.1:2, adding each component sequentially in this order.” .

- Were there incubation times in between the consecutive additions?

Yes. The incubation time is 30 min, and it has been added to the Methods section.

19. p15: “All crystals were cryoprotected using the corresponding reservoir solution supplemented with 20% glycerol or ethylene glycol...” .

- Which cryoprotectant for which crystal?

Crystals of apo-AcrIIA15, AcrIIA15-DNA and AcrIIA15^{NTD}-DNA were cryoprotected using the corresponding reservoir solution supplemented with 20% ethylene glycol, 20% ethylene glycol and 20% glycerol, respectively. These parameters have been added to the method.

20. p13, p16

- References for HKL3000, PHASER, SerialEM, CTFFIND4.

References for HKL2000, HKL3000, PHASER, SerialEM, and CTFFIND4 have been added to the revised manuscript.

21. p13: “The phasing model used for the structure of full-length AcrIIA15 was a AcrIIA15CTD structure which was solved previously using single-wavelength anomalous dispersion (SAD) method.” .

- Is this structure deposited in the PDB?

No.

22. p13: “To assemble the SaCas9-sgRNA-AcrIIA15CTD ternary complex, we mixed SaCas9, AcrIIA15CTD, and sgRNA sequentially at a molar ratio of 1:1.1:1.2.”

- This is counterintuitive. Why this order?

In the very beginning, we thought that AcrIIA15CTD might hinder the loading of sgRNA (no hindrance in fact as proved later in this manuscript). Hence, we added AcrIIA15CTD before sgRNA when assembling the ternary complex.

23. p13: “C322 Cu 300 mesh grids and C411 Cu 400 mesh grids …” .
- Quantifoil or C-flat grids?

Neither. C322 and C411 grids were purchased from a manufacture based in China.

24. p16: “an initial model was obtained and used for 3D classification …” .
-How was the initial model obtained?

After 2D classification in cryoSPARC, a subset of particles was selected to obtain an initial model using the program Ab-initio Reconstruction in cryoSPARC.

24. Fig1. It would have been nice to show the elution curve of AcrIIA15CTD to be able to compare it with that of the Cas9-sgRNA-DNA-AcrIIA15CTD complex.

There are excess AcrIIA15CTD component in the SaCas9-sgRNA-DNA-AcrIIA15CTD fraction, which is eluted around retention volume 16 ml indicated by an arrow in Fig. 1b. In addition, the elution peaks of free sgRNA and DNA are also indicated.

Reviewer #3 (Remarks to the Author):

Summary

In this manuscript, Deng et al. set out to characterize the molecular mechanisms of AcrIIA15 inhibition of SaCas9. Previous studies had suggested that AcrIIA15 acts by inhibiting sgRNA binding to SaCas9. In a series of biochemical and structural analyses, the authors refute this and tease apart the role of the AcrIIA15 N-terminal domain (NTD) in dimerization and promoter DNA binding, and the role of the C-terminal domain (CTD) in mimicking the target dsDNA to bind and inhibit SaCas9. The authors convincingly show that dimeric AcrIIA15 can simultaneously bind to promoter DNA inverted repeat (IR) regions via its NTD, and two SaCas9-sgRNA complexes via its CTD. In doing so, they have nicely investigated the structural and biochemical basis for the dual functions of AcrIIA15. This is a very strong paper which makes a great contribution to our understanding of AcrII family members and an intriguing domain fusion. However, certain experiments should be conducted to further strengthen the conclusions of this study, particularly around the statistical robustness of the biochemistry presented and the biological relevance of the findings described within the submission.

We thank the Referee for the positive feedback and the constructive comments.

Major points

1. Cleavage gels should be quantified (e.g., percent cleaved) and statistical analyses performed to assess whether the inhibition of cleavage is significant (Main Fig 1a, Fig 3g; Extended data Fig 1a-b, Fig 2). This is also relevant for binding/EMSA gels (Main Fig 4a, Fig 5e-f, Fig 6e; Extended data Fig 3, Fig 5, Fig 7e-f) which should be quantified (e.g., percent bound) and statistical analyses performed to assess whether differences in binding are significant. Including statements of how

many times these experiments were performed in each figure legend is also needed, with 3 independent experiments expected.

We thank the Referee for this suggestion. In the revision, each experiment was repeated at least three times. The bands of cleavage assays have been quantified and subjected to statistical analyses for data of Fig 1a, Fig 3g; Supplementary Fig. 2. Regarding to the binding/EMSA gels, since there are more than one bound band in some lanes and some bands are not sharp, it is not accurate to quantify the amount of binding. Hence, only representative gels were shown for figures of the EMSA assays.

2. Determining the affinities of the various interactions explored in this paper would strengthen this study (e.g., AcrIIA15 for SaCas9 versus target DNA for SaCas9; AcrIIA15 for IR1 and/or IR2). As a particular example, the authors suggest that the binding affinity between AcrIIA15 and SaCas9-sgRNA is lower when the WED domain interacting residue Y162 is mutated (Extended data Fig 4). This conclusion would be substantiated by quantifying and comparing the binding affinities of WT and mutant AcrIIA15 for SaCas9-sgRNA. Understanding the affinity of these interactions might also give additional insight into how AcrIIA15 functions. E.g., is the affinity of AcrIIA15 for the IR promoter DNA higher than the affinity between AcrIIA15 and SaCas9, or vice versa?

We thank the reviewer for these suggestions.

In response to the binding affinity concern, we conducted additional Isothermal Titration Calorimetry (ITC) experiments to assess the interactions between AcrIIA15 (wild-type and mutants), SaCas9-sgRNA, and target DNA. Despite our best efforts, obtaining precise K_d values proved challenging. The primary obstacle was the exceptionally high binding affinities, surpassing the optimal range for ITC measurements. This limitation hindered accurate quantification of binding constants for AcrIIA15, DNA, and Cas9-sgRNA. The extreme strength of these interactions posed challenges in obtaining reliable ITC data. Moreover, conformational changes in Cas9 during binding introduced unpredictability, leading to inaccuracies in the ITC results.

We compared the relative binding affinities of AcrIIA15 to SaCas9 and the target DNA to SaCas9 using SEC. When AcrIIA15 and the target DNA were added sequentially, both initial additions formed stable ternary complexes with the SaCas9-sgRNA binary complex, while later additions could not compete. However, simultaneous introduction of AcrIIA15 and target DNA resulted in minimal DNA binding, with the majority of AcrIIA15 binding to the SaCas9-sgRNA binary complex (Fig. R2). This observation suggests that AcrIIA15 has a higher binding affinity for SaCas9 than the target DNA.

Furthermore, to determine the dissociation constants of AcrIIA15 for IR1 and/or IR2, we performed SPR and SEC assays. Our findings indicate that full-length AcrIIA15 exhibits high binding affinity to IR1, IR2, and the entire IR1-IR2, at the nanomolar level. The SEC assay revealed that AcrIIA15 binds to the IR1-IR2 dsDNA sequence at a molar ratio of 4 to 1, suggesting simultaneous binding of two AcrIIA15 dimers to one IR1-IR2 DNA molecule. These data are presented in Supplementary Fig. 7a, b in the revised version.

Fig. R2 | Competition assay of DNA and AcrIIA15^{CTD} for binding to the SaCas9-sgRNA complex. Curve 1 (blue): sequential addition of AcrIIA15^{CTD} and DNA. Curve 2 (black): sequential addition of DNA and AcrIIA15^{CTD}. Curve 3 (red): simultaneous addition of AcrIIA15^{CTD} and DNA. Samples from major peaks, marked by two dotted vertical lines, underwent SDS-PAGE and Urea-PAGE for protein and nucleic acid content assessment.

3. In the context of AcrIIA15 activity during phage infection, AcrIIA15 would encounter a much larger stretch of dsDNA (indeed, an entire phage genome in some cases) where the Acr promoter region contains two IRs (IR1 and IR2). The authors speculate that both IRs could be simultaneously bound by two AcrIIA15 dimer:SaCas9 complexes. To strengthen this claim and increase the biological relevance of their study, the authors should attempt experiments to assess the stoichiometry of binding to the full-length promoter (IR1 + IR2) dsDNA sequence, at least by SEC.

We thank the reviewer for this constructive suggestion.

In the revised version, we performed SEC analysis to determine the binding ratio between AcrIIA15 dimers and (IR1-IR2) dsDNA sequences. The experiment involved incubating a constant amount of dsDNA with varying concentrations of AcrIIA15 at different molar ratios (1:2, 1:3, 1:4, and 1:6). Notably, at the 1:4 and 1:6 molar ratios, DNA and AcrIIA15 formed a single peak, indicative of a one DNA-two AcrIIA15 dimers complex. In contrast, at the 1:3 molar ratio, two peaks of DNA-AcrIIA15 complexes were observed, suggesting that one or two AcrIIA15 molecules bind to one DNA. Additionally, at the 1:2 molar ratio, a peak corresponding to one AcrIIA15 dimer-DNA complex was observed, along with free DNA. These findings collectively support the conclusion that one (IR1-IR2) dsDNA can bind to two AcrIIA15 dimers, establishing a stoichiometric ratio of AcrIIA15 to DNA as 4:1. Detailed data can be found in Supplementary Fig. 7b.

4. For all experiments involving point mutants of the AcrIIA15 protein (e.g. Main Fig 3g, Fig 5f, Fig 6e; Extended Data Fig 7e-f), performing analyses (such as DSF, CD, etc) to determine whether the mutant proteins retain a similar structure/fold to WT AcrIIA15 protein would strengthen the conclusion that these residues are important and the lack of inhibition/DNA binding is specific to the mutation and not due to misfolding.

We thank the reviewer for this suggestion. We have performed circular dichroism spectroscopy to

determine whether mutations have changed the folding of AcrIIA15. With the exception of R33A (Fig. R1), all other AcrIIA15 mutants involved in inhibition, DNA binding, and dimerization displayed CD spectra similar to WT AcrIIA15. This suggests that these mutations do not induce misfolding of AcrIIA15. These data are referred to Supplementary Fig. 4.

5. What is the functional consequence of the binding of both the IR in phage DNA and SaCas9? How does binding to one domain of AcrIIA15 affect subsequent binding to the other domain, and whether there is any allostery involved (e.g. does binding to SaCas9-sgRNA and conformational change affect the affinity of promoter DNA binding or vice versa). Does binding to the SaCas9-sgRNA complex affect transcriptional regulation of the Acr operon? The authors point to AcrIIA1 in the discussion as an example of a link between Cas9 abundance and the Acr operon. Could the authors test this experimentally in vivo using a reporter model that encodes GFP and include the constructs with and without NTD/CTD or SaCas9 to explore this?

The N-terminal domain (NTD) of AcrIIA15 binds to the upstream region of its own promoter, leading to self-repression. Meanwhile, the C-terminal domain (CTD) of AcrIIA15 binds to Cas9 inhibiting the DNA cleavage activity of SaCas9. The joint binding of both the inverted repeat (IR) and SaCas9 protects the phage genome from Cas9 cleavage and regulates its own expression levels, triggering the lysogenic cycle during the late stages of infection. AcrIIA1 shares a similar NTD domain with AcrIIA15, suggesting a potential common biological function for the helix-turn-helix (HTH) motif in these Acr proteins. The details of these binding interactions are discussed in our revised manuscript.

To investigate the potential impact of binding to one domain of AcrIIA15 on subsequent binding to the other domain, we conducted a structural comparison between the AcrIIA15-DNA complex and the SaCas9-sgRNA-AcrIIA15-DNA quaternary complex by aligning their DNAs. The structural alignment revealed a well-aligned NTD in both structures, while the CTD displayed distinct conformations (Fig. R3). We should note that the linker connecting the NTD and CTD domains of AcrIIA15 is a flexible loop. It indicates that the two CTD domains in a dimer can easily rotate outward, utilizing the linker as a hinge. This rotation allows the CTD domains to move away from each other, creating space to accommodate Cas9. Therefore, these findings suggest that the binding to one domain of AcrIIA15 does not significantly influence subsequent binding to the other domain.

Fig. R3 | Structural comparison of the AcrIIA15-DNA (gray) and SaCas9-sgRNA-AcrIIA15-DNA complexes (teal). Two views of the superimposed structures are displayed, and the cartoon

representation of SaCas9-sgRNA is omitted for clarity. The rotating directions of the CTDs of the two AcrIIA15 monomers upon binding to SaCas9 are indicated by arrows.

To further assess whether the binding to SaCas9-sgRNA affects the binding affinity of NTD to DNA inverted repeat 1 (IR1), we conducted an electrophoretic mobility shift assay (EMSA) using Cy3-labeled IR1 DNA. Both free AcrIIA15 and the SaCas9-sgRNA-AcrIIA15 complex were examined (Fig. R4). The results indicate that free AcrIIA15 exhibits a slightly higher binding affinity to IR1 compared to the SaCas9-sgRNA-bound AcrIIA15. This suggests that the binding of SaCas9-sgRNA to AcrIIA15 slightly diminishes the binding affinity between AcrIIA15 and IR1. Additionally, these findings imply that AcrIIA15 can sense the expression level of SaCas9 to finely regulate the transcription of its own gene.

Fig. R4. | The EMSA assay to assess binding affinities between free AcrIIA15 (left panel) or Cas9-sgRNA-bound AcrIIA15 (right panel) and IR1 DNA. In this assay, one strand of IR1 DNA was labeled with cy3 at the 5'-end. A gradient of AcrIIA15 concentrations ranging from 4 to 20 μ M was utilized, and DNA was kept to a constant concentration of 500 nM. For Cas9-sgRNA-bound AcrIIA15, the molar ratio of Cas9:sgRNA:AcrIIA15 is 1:1.1:0.8. The two bound IR1 bands correspond to AcrIIA15 dimer complexes with one or two Cas9-sgRNA.

Minor points

1. Figure 1: Would benefit from a schematic of the full-length vs CTD AcrIIA15 constructs to orient the reader (like what is shown in Fig 2a).

We thank the reviewer for this suggestion. The schematic of full-length vs CTD AcrIIA15 constructs have been incorporated into the revised Fig. 1a.

2. Figure 1a: Define FL as full length (FL) in the figure legend for clarity.

The term "FL" was replaced with "Full-length" in the revised version.

3. Figure 1: Inhibition of cleavage starts at a ratio of 1:10, with a ratio of 1:20-1:50 needed for a what appears to be a complete shift to uncleaved. This would suggest that the affinity is relatively

low. It is not clear from the method what the exact concentrations are in addition to the molar ratios indicated.

We appreciate the reviewer's suggestion. Full inhibition can be achieved at a molar ratio of 1:10 for AcrIIA15 and AcrIIA15CTD after increasing the salt concentration from 100 mM KCl to 300 mM KCl. This effect is likely attributed to the enhanced stability of AcrIIA15 in a higher-salt buffer. In the reaction, the concentrations of SaCas9 and sgRNA are 100 nM and 110 nM, respectively. The specific Cas9:Acr molar ratios are illustrated in the revised Figure 1. In addition, the method for enzymatic complex preparation has been amended in the Methods section of this revised manuscript.

4. Figure 1b: The authors should indicate the ratio and/or concentration of each component used in the complexing SEC experiments. E.g., was it 1:1 ratio? Based on the Coomassie gel, there appears to be a huge excess of SaCas9 compared to AcrIIA15-CTD in the major peak – was unbound CTD found in some of the later peaks?

We appreciate the reviewer's suggestion. The molar ratios of each component used in the complexing size-exclusion chromatography (SEC) experiments are now indicated in the revised figure legend.

In Figure 1b, the major peak does not show an excess of SaCas9 compared to AcrIIA15-CTD. The substantial difference in the intensity of the SaCas9 and AcrIIA15-CTD bands is primarily due to the significant contrast in their molecular weights. SaCas9 has a molecular weight of 124 kDa, while AcrIIA15-CTD has a molecular weight of 13.8 kDa, despite the apparent molecular weight of AcrIIA15-CTD appearing to be around 20 kDa.

The weak AcrIIA15CTD band in the absence of sgRNA is attributed to the poor binding affinity between SaCas9 and AcrIIA15-CTD. Conversely, the presence of sgRNA significantly enhances the binding affinity between AcrIIA15-CTD and SaCas9, resulting in a more pronounced band for AcrIIA15-CTD. In both runs, there is an excess of AcrIIA15-CTD, as indicated in the revised Figure 1b.

5. The authors should comment on the observed specificity of AcrIIA15 for inhibition of SaCas9 – are there any clues as to how this specificity is achieved, given all three Cas9 orthologs bind dsDNA and would presumably be inhibited by DNA-mimicking AcrIIA15? Or does this reflect differences in the PAM motif recognised by the different orthologs? Are the AcrIIA15-interacting residues conserved amongst Cas9 orthologs? A small multiple sequence alignment of Cas9 orthologs could answer this question and be an interesting insight into AcrIIA15 specificity. This could be brought up in the discussion.

We appreciate the reviewer's suggestion. The sequence alignments of three Cas9 orthologs (SaCas9, SpCas9, and Nme1Cas9) reveal that the residues interacting with AcrIIA15 are not conserved. This observation provides insights into the specificity of AcrIIA15, explaining why it specifically inhibits SaCas9 but does not affect SpCas9 and Nme1Cas9. Additionally, sequence alignments also show that PAM recognition residues are not conserved at all. Since the PAM recognition residue N985 of

SaCas9 is critical for AcrIIA15 inhibition (Fig. 3b, g), the specificity of AcrIIA15 does reflect differences in PAM motif. In the revised manuscript, the sequence alignments of the three Cas9s have been included in Supplementary Fig. 5.

6. Figure 2: minor suggestion – clarity might be improved if the Cas9 domains were coloured in blue/green/cool tones, and the ACR domains in warm/pink tones or maybe even red to match the DNA (the pink Cas9 HNH domain is close in colour to the ACR NTD). Likewise, might be good to colour the B-form DNA in red in 2e. It would be good to define the acronyms used for different lobes and domains, if not in the figure legend then somewhere in the text particularly for a non-expert reader.

We appreciate the reviewer's suggestion. After comparing the current colors with the suggested colors for the overall structure of Cas9-sgRNA-AcrIIA15, we found that the existing colors for each domain remain as clear as the suggested. Therefore, we have decided to retain the current colors. Additionally, we have defined acronyms for different lobes and domains in the manuscript in accordance with the suggestions.

7. Figure 3: 3g: Would be good to have some indication of ratio/concentration used here. Was it 1:50 to achieve max WT inhibition? And was the ratio/concentration kept consistent for each mutant – important to state this.

The ratio of Cas9 to Acr used in Fig. 3g is 1:10, and this ratio, along with the concentrations, was maintained consistently for each mutant. The details on the ratios and concentrations have been included in the figure legend of Fig. 3g in the revised version.

8. Figure 4: 4a: At lower ratios DNA:Acr (e.g. 1:1 and 1:2), it seems that the IR1 is bound first (smaller band appears first) before the IR2 is bound (larger band appears from 1:2 onward). Why are the Acr-bound IR1 and IR2 bands two different sizes when the input DNA appears to be the same size (single band in the DNA only lane)? Was the ratio/concentration of IRI1 and IRI2 the same in the mixed experiment (4a left most)?

We thank the reviewer for raising this concern. The apparent difference in size between the Acr-bound IR1 and Acr-bound IR2 bands is solely due to the varying scale of these panels. We have made revision to clarify this aspect in the figure. For the Fig. 4a leftmost experiment, the DNA (IR1-IR2) is virtually a single dsDNA harboring both IR1 and IR2. The two bound bands correspond to (IR1-IR2) dsDNA bound to one or two AcrIIA15 dimers.

9. Do the IR regions (similar to IR1 and IR2) occur elsewhere in the phage genome? Are other genes possibly regulated by AcrIIA15 binding?

No.

10. Figure 5: 5f: it was unclear from the figure annotations and figure legend whether these point mutations were introduced into the full length AcrIIA15 protein or in the AcrIIA15-NTD protein.

The introduced point mutations were incorporated into the full-length AcrIIA15 proteins. This clarification has been emphasized in the figure legend of Fig. 5f.

11. Extended data Fig 5: When assessing ability of AcrII15-NTD to bind IR1:IR2 by EMSA, we see a very different band shift pattern compared to full length AcrII15, in particular multiple 'bound' bands which increase in shift as amount of AcrII15-NTD is increased. Could the authors comment on this?

In the revised version, the Extended Data Fig 5 is now represented as Supplementary Fig. 7c. The AcrIIA15-NTD is not stable and may polymerize at high protein concentrations, leading to a slower band shift, particularly evident in the 1:10 and 1:20 molar ratios. In the revision, we omitted the lanes corresponding to these ratios from the EMSA to avoid any potential misinterpretation.

12. Extended data Fig 6B – The authors included extra bp outside of the IR1 inverted repeats in order for DNA:AcrII complex to be achieved. Does this imply that the bp outside the IR are also critical for DNA binding? And why was the decision made to have 1 bp overhangs in DNA2, why not dsDNA?

Thank you for raising this point. In the revised version, the Extended Data Fig 6B is now Supplementary Fig. 8b. It's important to note that the additional base pair highlighted by the reviewer is situated inside IR1, rather than outside the IR.

In our efforts to determine the minimal length of IR1 required for NTD domain binding, we truncated IR1 from both ends. Notably, a 4-bp truncation from both ends (DNA1, designed as IR1' in the revised manuscript) significantly decreased the binding affinity between IR1 and AcrIIA15. However, the addition of one nucleotide on the 5'-end of both strands (DNA2, designed as IR1'' in the revised manuscript) remarkably restored binding. This observation emphasizes the critical role of the central 20-bp within IR1 for AcrIIA15 binding. We revised this section for clarity in the revision.

The 1-bp overhangs in DNA2 (IR'') facilitated the crystallization. To achieve high-quality crystals, we utilized DNA2 (IR''), incorporating a 1-bp overhang, in contrast to dsDNA with a blunt end.

REVIEWERS' COMMENTS

Reviewer #2 (Remarks to the Author):

The authors have thoroughly addressed all the comments and performed additional experiments that strengthen and improve the paper.

Reviewer #3 (Remarks to the Author):

Deng et al have addressed our initial set of comments effectively and performed the additional experiments that were requested. Overall the data they now include have been appropriately described and provide the critical context for the work. With a few minor changes we believe this manuscript is suitable for publication (subject to the editor's determination of with respect to the journal scope and associated standards).

Final Revisions:

- 1) Methods section describing cleavage gel quantification and what statistical analysis (e.g. ANOVA, t-test, etc)
- 2) Line 111: "sever" should be "server"
- 3) Lines 179-180: "high affinity" is mentioned twice in this sentence, remove one mention

Manuscript NCOMMS-23-36510A

‘An anti-CRISPR that represses its own transcription while blocking Cas9-target DNA binding’

We thank the reviewers for their encouraging comments and critical feedback. Their suggestions are very helpful. Our point-by-point responses (in blue) to the reviewers’ comments (in black) are as below:

REVIEWERS' COMMENTS

Reviewer #2 (Remarks to the Author):

The authors have thoroughly addressed all the comments and performed additional experiments that strengthen and improve the paper.

We thank the reviewer for the support.

Reviewer #3 (Remarks to the Author):

Deng et al have addressed our initial set of comments effectively and performed the additional experiments that were requested. Overall the data they now include have been appropriately described and provide the critical context for the work. With a few minor changes we believe this manuscript is suitable for publication (subject to the editor’ s determination of with respect to the journal scope and associated standards).

Final Revisions:

1) Methods section describing cleavage gel quantification and what statistical analysis (e.g. ANOVA, t-test, etc)

The description has been illustrated in a new Method section “Statistics and Reproducibility”.

2) Line 111: “sever” should be “server”

The typo has been corrected.

3) Lines 179-180: “high affinity” is mentioned twice in this sentence, remove one mention

The phrase “with high affinity” in the sentence “We found that the full-length AcrIIA15 exhibits high binding affinity ……” has been deleted in the revised manuscript.